



Atmospheric
Chemistry
and Physics

# Measurement report: Source apportionment of volatile organic compounds at the remote high-altitude Maïdo observatory

**Bert Verreyken**[1,2,3], **Crist Amelynck**[1,2], **Niels Schoon**[1], **Jean-François Müller**[1], **Jérôme Brioude**[3], **Nicolas Kumps**[1], **Christian Hermans**[1], **Jean-Marc Metzger**[4], **Aurélie Colomb**[5], and **Trissevgeni Stavrakou**[1]

[1]Royal Belgian Institute for Space Aeronomy, 1180 Brussels, Belgium
[2]Department of Chemistry, Ghent University, 9000 Ghent, Belgium
[3]Laboratoire de l'Atmosphère et des Cyclones, UMR 8105, CNRS, Université de La Réunion, Météo France, 97744 Saint-Denis, France
[4]Observatoire des Sciences de l'Univers de La Réunion, UMS3365, CNRS, Université de La Réunion, Météo-France, Saint-Denis, La Réunion, France
[5]Laboratoire de Météorologie Physique, UMR 6016, CNRS, Université Clermont Auvergne, 63178 Aubière, France

**Correspondence:** Bert Verreyken (bert.verreyken@aeronomie.be)

**Abstract.** We present a source apportionment study of a near-continuous 2-year dataset of volatile organic compounds (VOCs), recorded between October 2017 and November 2019 with a quadrupole-based high-sensitivity proton-transfer-reaction mass-spectrometry (hs-PTR-MS) instrument deployed at the Maïdo observatory (21.1° S, 55.4° E, 2160 m altitude). The observatory is located on La Réunion island in the southwest Indian Ocean. We discuss seasonal and diel profiles of six key VOC species unequivocally linked to specific sources – acetonitrile ($CH_3CN$), isoprene ($C_5H_8$), isoprene oxidation products (Iox), benzene ($C_6H_6$), $C_8$-aromatic compounds ($C_8H_{10}$), and dimethyl sulfide (DMS). The data are analyzed using the positive matrix factorization (PMF) method and back-trajectory calculations based on the Lagrangian mesoscale transport model FLEXPART-AROME to identify the impact of different sources on air masses sampled at the observatory. As opposed to the biomass burning tracer $CH_3CN$, which does not exhibit a typical diel pattern consistently throughout the dataset, we identify pronounced diel profiles with a daytime maximum for the biogenic ($C_5H_8$ and Iox) and anthropogenic ($C_6H_6$, $C_8H_{10}$) tracers. The marine tracer DMS generally displays a daytime maximum except for the austral winter when the difference between daytime and nighttime mixing ratios vanishes. Four factors were identified by the PMF: background/biomass burning, anthropogenic, primary biogenic, and secondary biogenic. Despite human activity being concentrated in a few coastal areas, the PMF results indicate that the anthropogenic source factor is the dominant contributor to the VOC load (38 %), followed by the background/biomass burning source factor originating in the free troposphere (33 %), and by the primary (15 %) and secondary biogenic (14 %) source factors. FLEXPART-AROME simulations showed that the observatory was most sensitive to anthropogenic emissions west of Maïdo while the strongest biogenic contributions coincided with air masses passing over the northeastern part of La Réunion. At night, the observatory is often located in the free troposphere, while during the day, the measurements are influenced by mesoscale sources. Interquartile ranges of nighttime 30 min average mixing ratios of methanol ($CH_3OH$), $CH_3CN$, acetaldehyde ($CH_3CHO$), formic acid (HCOOH), acetone ($CH_3COCH_3$), acetic acid ($CH_3COOH$), and methyl ethyl ketone (MEK), representative for the atmospheric composition of the free troposphere, were found to be 525–887, 79–110, 61–101, 172–335, 259–379, 64–164, and 11–21 pptv, respectively.

# 1   Introduction

Non-methane volatile organic compounds (NMVOCs) are key players in atmospheric chemistry. Their reaction with the main atmospheric oxidants modulates the oxidative capacity of the atmosphere (Zhao et al., 2019) and, in combination with $NO_x$, results in the production of tropospheric ozone ($O_3$) and secondary organic aerosol (SOA), both air pollutants affecting human health (e.g., Jerrett et al., 2009) and short-term climate forcers. Whereas increases in tropospheric $O_3$ contribute to global warming, SOA affects radiative forcing through both direct interaction with radiation and indirectly acting as cloud condensation nuclei (CCN), resulting in an overall cooling effect (IPCC, 2013). Besides pure hydrocarbon compounds, which are emitted in huge amounts at the global scale and mainly by terrestrial vegetation, an important NMVOC class is composed of oxygenated compounds (OVOCs). These OVOCs can be directly emitted to the atmosphere by multiple sources, including the biosphere (vegetation and soils), biomass burning, anthropogenic activities, and the ocean. Secondary production by oxidation of primary emitted NMVOCs, however, is a very important and often badly quantified source for many OVOCs as well. Moreover, recent research has pointed to the importance of bidirectional exchange of OVOCs with oceans (Yang et al., 2014) and terrestrial vegetation (Farmer and Riches, 2020), a process which is not well described in atmospheric models and which complicates OVOC budget calculations. In the absence of high concentrations of highly reactive non-methane hydrocarbons, OVOCs are expected to account for most of the OH reactivity in the remote tropical marine atmosphere (Travis et al., 2020). However, due to scarcity of observational constraints, OVOC sources and sinks are prone to large uncertainties in these regions (Millet et al., 2010; Read et al., 2012; Travis et al., 2020). The OCTAVE project (https://octave.aeronomie.be, last access: 11 December 2020) aims at reducing those uncertainties through in situ measurements, satellite retrievals of global OVOC concentrations, and tropospheric modeling. In the framework of this project, a quadrupole-based high-sensitivity proton-transfer-reaction mass-spectrometry VOC analyzer (hs-PTR-MS) was deployed for 2 years at the remote high-altitude Maïdo observatory (21.1° S, 55.4° E, 2160 m altitude) at La Réunion island, a remote tropical French volcanic island in the southwest Indian Ocean. The instrument continuously measured marine boundary layer air enriched with compounds originating from urbanized areas and ecosystems native to the island during the day. At night, the observatory is frequently located in or near the free troposphere. Part of this hs-PTR-MS dataset (April 2018), in combination with Aerolaser formaldehyde measurements, has already been used for formaldehyde source apportionment on the island using positive matrix factorization (PMF) (Rocco et al., 2020). Another study focused on the detection of African biomass burning plumes during August 2018 and 2019 and their impact on the

(O)VOC composition at the Maïdo observatory (Verreyken et al., 2020).

The present study makes use of the complete dataset and aims at a better characterization of mesoscale (O)VOC sources by studying their seasonal, diel, and inter-annual variability using PMF and back-trajectory calculations. Previous studies have described the mesoscale transport features over La Réunion and the impact on measurement campaigns taking place on the island (Lesouëf et al., 2011; Baray et al., 2013; Tulet et al., 2017; Guilpart et al., 2017; Foucart et al., 2018; Duflot et al., 2019). The recent development of FLEXPART-AROME (Verreyken et al., 2019) – a Lagrangian transport model driven by meteorological data obtained from the operational mesoscale numerical weather prediction models used in the region by Météo France – provides the opportunity to study mesoscale transport and its impact on the near-continuous 2-year (O)VOC dataset recorded at the observatory. We start by describing the measurement site, the instruments, and the source attribution tools used in Sect. 2. We present the diel, seasonal, and inter-annual variability of key tracers linked to known sources in Sect. 3.1. This variability is discussed in light of sources identified by the PMF algorithm (Sect. 3.2) and back-trajectories calculated with FLEXPART-AROME (Sect. 3.3).

# 2   Materials and methods

## 2.1   Description of measurement site

La Réunion – a small tropical island located in the southwest Indian Ocean, shown in Fig. 1 – is home to the high-altitude Maïdo observatory (21.1° S, 55.4° E, 2160 m altitude) (Baray et al., 2013). Despite urbanization of the coastal areas, the island still has about 10 000 ha of native ecosystems (Duflot et al., 2019). The Maïdo observatory is located in a national park and surrounded by mountain shrublands and heathlands (Duflot et al., 2019). The largest city is the capital, Saint-Denis. Other large cities (population over 50 000) are Saint-Paul, Saint-Pierre, and Le Tampon. Industrial emission hotspots at La Réunion are located near the power plants in Le Port (diesel power plant), Le Gol (biomass power plant), and Bois-Rouge (biomass power plant). The island is home to the very active Piton de La Fournaise volcano (Tulet et al., 2017). La Réunion is isolated from large landmasses and provides an ideal location to study complex processes and interactions in the remote atmosphere of anthropogenic, biogenic, and volcanic emissions as illustrated by studies of e.g., volcanic plume emissions and aging (Tulet et al., 2017), new particle formation (Foucart et al., 2018), and forest–gas–aerosol–cloud system interaction (Duflot et al., 2019). A recently published land cover map of La Réunion (Dupuy et al., 2020) shows that a large fraction of the coastal zone is used to cultivate sugar cane, which is by far the dominant agricultural crop, and fruit. The eastern part of the island is mainly

**Table 1.** Special events with potential impact on VOC concentrations during the 2-year deployment of the hs-PTR-MS. During some of these events the hs-PTR-MR was shut down due to security measures at the Maïdo observatory. Information is from Fournaise Info (2018), Le Monde (2018), and Météo France (2020).

| Year | Start | End | Duration | Type | Comments |
|------|-------|-----|----------|------|----------|
| 2017 | 27 December | 9 January | 13 d | Tropical cyclone, Ava | Technical issues with the hs-PTR-MS instrument |
| 2018 | 9 January | 20 January | 11 d | Intense tropical cyclone, Berguitta | Maïdo closed |
| | 1 March | 6 March | 5 d | Intense tropical cyclone, Dumazile | Maïdo closed |
| | 13 March | 20 March | 7 d | Severe tropical storm, Eliakim | |
| | 3 April | 4 April | 17 h | Volcanic eruption | |
| | 20 April | 25 April | 5 d | Tropical cyclone, Fakir | |
| | 27 April | 25 May | 27 d | Volcanic eruption | Accompanied by occasional vegetation fires |
| | 13 July | 13 July | 13 h | Volcanic eruption | |
| | 17 November | 25 November | 8 d | Social unrest due to the "yellow vests" protests | Last 5 d had a partial curfew |
| 2019 | 18 February | 10 March | 21 d | Volcanic eruption | |
| | 10 June | 13 June | 54 h | Volcanic eruption | |
| | 29 July | 30 July | 24 h | Volcanic eruption | |
| | 11 August | 15 August | 4 d | Volcanic eruption | |
| | 25 October | 27 October | 2 d | Volcanic eruption | |

covered by woodlands. The western region is typically drier throughout the year and has relatively more shrub and herbaceous savanna near the coast (Dupuy et al., 2020). Orographically induced precipitation along the western mountain slope allows woodlands to be dominant at mid-level altitudes (Dupuy et al., 2020). There are two distinct seasons on the island, the warm and wet season (December to March) and the cold, dry season (May to November) (Foucart et al., 2018). From October to May, the region is sensitive to tropical cyclone activity. Synoptic-scale air-mass transport in the region is dominated by east-southeasterly trade winds near the surface and westerlies in the free troposphere (FT) (Baldy et al., 1996; Lesouëf et al., 2011; Baray et al., 2013). These trade winds weaken from December to March but intensify from April to November (Baldy et al., 1996). The complex orographic profile of La Réunion (highest point over 3000 m altitude) introduces a major obstacle in the stable wind flow pattern. Trade winds are split around the island, with winds accelerating along the coastlines parallel to the synoptic flow (Lesouëf et al., 2011). The northwestern area (lee side) of the island is sheltered from trade winds by the mountainous profile. Counter-flowing vortices in the wake of the island can trap polluted air masses (Lesouëf et al., 2011). Transport in

the northwestern sector of the island is dominated by the coupling of sea (land)–breeze with upslope (downslope) transport during the day (night). During the day, the Maïdo observatory – located west of the Maïdo mountain peak – resides in the planetary boundary layer (PBL) while at night it is frequently in or near the FT (Lesouëf et al., 2011; Baray et al., 2013; Guilpart et al., 2017; Duflot et al., 2019). During the day, a horizontal wind shear front located at the confluence of the mesoscale-driven upslope transport and overflowing trade winds determines air-mass origins at the observatory (Duflot et al., 2019). When the front is west of the observatory, surface emissions have less impact on the composition of air masses as they originate mostly from 2000 m altitude (Duflot et al., 2019). During the OCTAVE measurement period, several events occurred which could have affected in situ air-mass composition and VOC mixing ratio diel profiles (tropical storms and cyclones, volcanic eruptions). These are summarized in Table 1 but are not studied in detail as their effects are outside the scope of the current work.

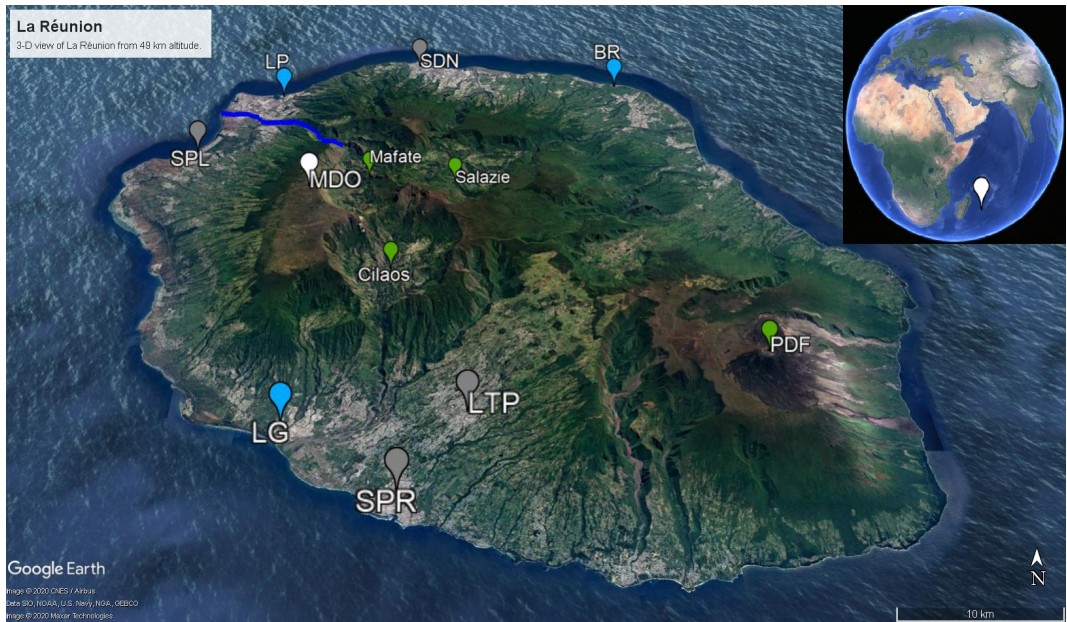

**Figure 1.** La Réunion, viewed from 49 km height. The Maïdo observatory (MDO) is indicated in white. Cities with population over 50 000 habitants – i.e., Saint-Denis (SDN), Saint-Paul (SPL), Le Tampon (LTP), and Saint-Pierre (SPR) – are shown in gray. The largest industrial sites – Le Port (LP), Bois-Rouge (BR), and Le Gol (LG) – are indicated in blue. Geological landmarks – Calderas Mafate, Salazie, and Cilaos as well as the shield volcano Piton de la Fournaise (PDF) – are in green. The river Rivière des Galets between Mafate and Le Port is indicated as a dark blue line. The image was generated with Google Earth Pro, data SOI, NOAA, U.S. Navy, GA, GEBCO. © 2020 CNES/Airbus, © 2020 Maxar Technologies.

## 2.2 Instrumentation and sampling setup

### 2.2.1 hs-PTR-MS

In October 2017, a high-sensitivity quadrupole-based proton-transfer-reaction mass-spectrometry instrument (hs-PTR-MS, Ionicon Analytik GmbH, Austria) was deployed at the Maïdo observatory. It was run in multiple ion detection mode to generate a near-continuous dataset of (mostly) oxygenated volatile organic compounds from 19 October 2017 to 26 November 2019. The instrument was operated in $H_3O^+$ reactant ion mode at a drift tube pressure and temperature of 2.2 hPa and 333 K, respectively, and at a drift field of 600 V, which resulted in an $E/N$ value (the ratio of the electric field to the number density in the drift tube) of 136 Td. VOC-related product ion signals were sequentially recorded at mass-to-charge ratios ($m/z$) of 31 (formaldehyde, HCHO), 33 (methanol, $CH_3OH$), 42 (acetonitrile, $CH_3CN$), 45 (acetaldehyde, $CH_3CHO$), 47 (formic acid, HCOOH), 59 (acetone, $CH_3COCH_3$), 61 (acetic acid, $CH_3COOH$), 63 (dimethylsulfide, DMS), 69 (isoprene, $C_5H_8$), 71 (methyl vinyl ketone, MVK; methacrolein, MACR; and possible contributions from isoprene hydroxy hydroperoxides (Rivera-Rios et al., 2014), ISOPOOH), 73 (methyl ethyl ketone, MEK), 79 (benzene, $C_6H_6$), 81 (sum of monoterpenes, $C_{10}H_{16}$), 93 (toluene, $C_7H_8$), 107 ($C_8$-aromatic compounds, $C_8H_{10}$), and 137 ($C_{10}H_{16}$), each with

a 10 s dwell time. In addition, ions at $m/z$ 21 ($H_3^{18}O^+$), 32 ($O_2^+$), and 37 ($H_3O^+ \cdot H_2O$) were monitored with dwell times of 2 s, 100 ms, and 100 ms, respectively. A complete measurement cycle lasted 2.7 min. Ion signals at $m/z$ 31, 81, and 93 were not considered in the present analysis because of potential contributions from interfering compounds or background ion signals from zero measurements which were considerably larger than the minimum values in ambient air, thus resulting in negative concentrations (specifically at $m/z$ 93). It is well documented that hs-PTR-MS measurements of MVK+MACR ($m/z = 71$) include contributions from ISOPOOH through the formation of MVK or MACR from ISOPOOH in the instrument (Liu et al., 2013; Rivera-Rios et al., 2014; Bernhammer et al., 2017). As ISOPOOH is the major first-generation oxidation product of $C_5H_8$ in low-nitrogen-oxide ($NO_x = NO + NO_2$) environments (Wennberg et al., 2018), we expect that the MVK+MACR signal could suffer from interference from ISOPOOH. Although MVK and MACR may have emissions related to anthropogenic activity (e.g., Biesenthal and Shepson, 1997), or biomass burning (e.g., Hatch et al., 2015), those are not expected to contribute significantly to the VOC composition at the location of the Maïdo observatory. Therefore, we will refer to ion signal at $m/z$ 71 as isoprene oxidation products or Iox. The ion signal at $m/z$ 47 may be the result of ion species of ethanol or HCOOH (Baasandorj et al., 2015). However, calibration of the instrumental

setup using a calibration gas unit provided by the Laboratoire des Sciences du Climat et de l'Environnement (LSCE, Climat and Environment Sciences Laboratory), which contains ethanol, showed a very low calibration coefficient for ethanol. Additionally, large ion signals recorded during periods when the observatory was under the influence of biomass burning events located in southern Africa and Madagascar (Verreyken et al., 2020) suggested that these signals were mainly due to HCOOH. Potential interference at $m/z$ 61 may originate from ambient concentrations of glycolaldehyde, ethyl acetate, and peroxyactic acid (Baasandorj et al., 2015). However, the strong co-variation between $CH_3COOH$ and HCOOH, which have similar sources and sinks in the atmosphere, suggests that the signal at $m/z$ 61 could correspond mostly to $CH_3COOH$. Air was sampled 2.86 m above the roof of the observatory (8.20 m above ground) and pumped towards the instrument through a 10.35 m long 3/8 in. (outer diameter) perfluoralkoxy (PFA) Teflon sampling tube (Dyneon 6502T, Fluortechnik-Deutschland, Germany) at a flow rate of 8 L min$^{-1}$. The sampling line was heated (5–10° above ambient temperature) and thermally insulated to prevent condensation. The sampled air was filtered for particles 2.48 m downstream of the sampling point by a polytetrafluoroethylene (PTFE) membrane filter (Zefluor, Pall Laboratory, MI, USA) with a pore size of 2 µm. Part of the sampled air was sent through a catalytic converter (type HPZA-3500, Parker Hannifin Corp., OH, USA) for zero-VOC measurements, which took place every 4 h and lasted 30 min, of which only the last minutes were taken into account. The transport time of ambient air between the sampling point and the drift tube reaction of the hs-PTR-MS instrument is about 3.2 s (2.5 s in the main sampling line and 650 ms in the hs-PTR-MS sampling line). Calibration of the hs-PTR-MS for the target VOCs was performed every 3–4 d by dynamically diluting a VOC / $N_2$ calibration mixture (Apel-Riemer Environmental Inc., FL, USA; stated accuracy of 5 % ($2\sigma$) on the VOC mixing ratios) in zero air. In April 2018 and March 2019, the instrument was also calibrated with a commercial gas calibration unit (GCU, Ionicon Analytik GmbH, Austria) from LSCE. Calibration factors for the (O)VOCs of interest obtained with the two calibration systems were found to be in excellent agreement. Calibration factors for compounds that were not present in the calibration mixture – i.e., HCOOH and $CH_3COOH$ – were determined indirectly, as mentioned in Verreyken et al. (2020). Specifically, for $CH_3COOH$, the calibration factor was inferred from that of $CH_3COCH_3$ by considering the calculated collision rate constants of $H_3O^+$ with $CH_3COOH$ and $CH_3COCH_3$ (Su, 1994; Zhao and Zhang, 2004), by considering the contribution of the protonated molecules to the respective product ion distributions (Schwarz et al., 2009; Inomata and Tanimoto, 2010), and by assuming the same hs-PTR-MS transmission efficiency for ions with a mass difference of 2 Da. The same principle was applied to calculate the calibration factor of HCOOH from that of $CH_3CHO$. The humidity dependence

of the calibration factors was determined approximately every 2 months by controlling the humidity of the zero air with a dew point generator (LI-COR LI-610, NE, USA). The ion signal at $m/z$ 37 was used as a proxy for air humidity. Of all compounds present in the calibration mixture, only the calibration factors for formaldehyde, isoprene, Iox, and MEK showed a non-negligible humidity dependence. The humidity dependence of calibration factors for carboxylic acids – not present in the calibration mixture – was retrieved from Baasandorj et al. (2015) for HCOOH and from the experimentally determined humidity-dependent fractional contribution of protonated $CH_3COOH$ to the $H_3O^2$ / $CH_3COOH$ product ions.

### 2.2.2 Additional measurements

The Maïdo observatory was recently officially registered as an ICOS (Integrated Carbon Observation System, https://www.icos-cp.eu/, last access: 7 January 2021) atmospheric measurement site and a GAW (Global Atmospheric Watch, https://public.wmo.int/en/programmes/global-atmosphere-watch-programme, last access: 7 January 2021) station. In this capacity, the observatory continuously houses a suite of both in situ and remote sensing instruments. A list of all regular measurements can be found online (https://osur.univ-reunion.fr/observations/osu-r-stations/opar/, last access: 11 December 2020). In this study we will focus on carbon monoxide (CO) mixing ratios taken by a Picarro G2401 instrument (Picarro Inc., CA, USA). We will use the CO data to better characterize the presence of biomass burning (BB) plumes at Maïdo. Auxiliary data (wind direction, ambient temperature, and solar radiation) were recorded by a meteorological station. The measurements of radiation have been made using a SPN1 Sunshine pyranometer (Delta-T Devices Ltd., UK), with a stated accuracy of 5 % for both direct and diffuse radiation. Mixing ratios of $NO_x$ have been recorded using a chemiluminescence photometer (T200UP, Envicontrol, France).

### 2.3 Source attribution tools

### 2.3.1 Positive matrix factorization

Air composition was studied using the positive matrix factorization (PMF) multivariate receptor model software released by the United States Environmental Protection Agency (EPA), EPA PMF 5.0 (Norris et al., 2014). PMF is a popular tool in atmospheric source attribution studies (e.g., Rocco et al., 2020; Pernov et al., 2021). The mathematical principle behind the PMF algorithm is based on the decomposition of measurements ($x_{ij}$) in a linear combination of factor profiles ($f_{kj}$) and factor contributions ($g_{ik}$) and a residual ($\epsilon_{ij}$):

$$x_{ij} = \sum_{k=1}^{p} g_{ik} \times f_{kj} + \epsilon_{ij}. \tag{1}$$

The indices $i$, $j$, and $k$ denote the measurement time, the measured compound, and the selected factor, respectively. The total number of factors ($p$) represents the number of sources affecting the dataset and is a hyperparameter, i.e., a parameter set by the analyzer to optimize the solution, for the algorithm. The only mathematical constraint to solve this equation is that all factor contributions and profiles must be positive. As a result, atmospheric sinks are not taken explicitly into account when deconstructing the dataset. The equation is solved by minimizing the objective function $Q$:

$$Q = \sum_{i=1}^{n} \sum_{j=1}^{m} \left[ \frac{\epsilon_{ij}}{u_{ij}} \right]^2, \qquad (2)$$

where $n$ is the total number of measurements, $m$ is the number of species, and $u_{ij}$ is the uncertainty of a measurement. In total, there are three different calculations of the objective function. The first takes into account all the residuals ($Q_{\text{true}}$), the second excludes a number of data points that are identified by the software as outliers ($Q_{\text{robust}}$), and the last version of the objective function is equal to the difference between the number of data values that are characterized as "strong" (see Sect. 3.2) and the number of parameters fitted by the model ($Q_{\text{expected}}$). The $Q_{\text{expected}}$ corresponds roughly to the number of degrees of freedom for the algorithm. The number of factors is set by scanning the parameter space and looking for a shift in both the $Q_{\text{true}}/Q_{\text{expected}}$ ratio – also called $Q_{\text{scaled}}$ – and the maximum root mean square (rms) of residuals for the different compounds as well as ensuring physical interpretability of the resulting factor profiles and contributions.

### 2.3.2 Back-trajectory modeling

The FLEXPART-AROME model, a limited-domain version of the Lagrangian transport model FLEXPART (Stohl et al., 2005; Pisso et al., 2019), was developed to simulate mesoscale transport over the complex orographic profile of La Réunion (Verreyken et al., 2019). The model is driven by meteorological data generated by AROME, the operational mesoscale numerical weather prediction model with a 2.5 km horizontal resolution used in the region by Météo-France. FLEXPART-AROME was initially developed from FLEXPART-WRF (Brioude et al., 2013) to forecast the dispersion of a volcanic plume for the STRAP campaign in 2015 (Tulet et al., 2017). However, as turbulence in the FLEXPART model is simulated using an independent parametrization from the numerical weather prediction model, inconsistencies in, e.g., PBL top definition may lead to unrealistic transport features in the offline transport model (Verreyken et al., 2019). In order to ensure harmonized turbulent transport in the numerical weather prediction model and the offline transport model, turbulence in FLEXPART-AROME was adapted to be driven by the 3-D turbulent kinetic energy fields obtained from AROME (Verreyken et al., 2019). The

FLEXPART-AROME model has been used to study impact of mesoscale transport on BB plumes from distant sources (Verreyken et al., 2020). The model is driven by combining AROME forecasts generated daily at 00:00, 06:00, 12:00, and 18:00 UT. From 3 November 2017 until 26 November 2019, 20 000 air parcels were initialized every hour between 0 and 20 m above ground level (a.g.l.) at 21.081° S, 55.383° E. The model is run in backward mode (Seibert and Frank, 2004). Air parcels are traced 24 h backward in time and are separated into two age classes of 12 h. Air parcels' residence times (RT, expressed in seconds) are computed in a grid with 0.025° (about 2.5 km) horizontal resolution between 19.5–22.5° S and 53.0–58.0° E with 15 vertical layers of 50 m thickness below 500 m a.g.l, 500 m thickness up to 2000 m a.g.l, and two additional layers above (10 and 24 km height). Output is generated every hour and contains the residence times; i.e., the accumulated time air masses initialized at a certain time $t$ are present in a certain grid cell, $m$, between two consecutive time steps, $l-1$ and $l$. The residence time is proportional to the impact of the emission rate of a source located in grid cell $m$ at time $l$ on the measurement related to the release of air parcels at the location of the observatory (receptor) in the model. This proportionality is quantified by the source–receptor relationship (SRR, also called emission sensitivity) of area $m$ (source) at time $l$ and is calculated by

$$\text{SRR}_{lm} = \frac{\text{RT}_{lm}}{h}, \qquad (3)$$

where $h$ is the height of the surface layer at time $l$. In practice, $h$ cannot be higher than the PBL but may not be too shallow in order to be numerically robust (Seibert and Frank, 2004). Concentrations at the receptor site of a passive tracer $X$ (not produced or lost during transport) can be calculated by

$$C_X = \sum_l \sum_m E_{X,lm} \times \text{SRR}_{lm}, \qquad (4)$$

where $E_{X,lm}$ is the emission rate (expressed in $\text{g m}^{-2} \text{s}^{-1}$) of tracer $X$ related to area $m$ at time $l$. As the (O)VOC compounds recorded with the hs-PTR-MS instrument are generally reactive compounds, the SRRs from back-trajectory calculations are compared to the more robust PMF source factor contribution temporal variations. This method can be used to estimate the impact of emissions on in situ measurements or to estimate the emission rates using an inverse modeling approach (e.g., Brioude et al., 2011). In this work, we will discuss the SRRs directly in order to describe the sensitivity of in situ measurements performed at the observatory to emissions on the island rather than go into detail on emission rates of specific sources. In this work, we will simulate the mesoscale boundary layer movement towards Maïdo using a minimal static PBL proxy used by Lesouëf et al. (2011). Lesouëf et al. (2011) studied the PBL development around La Réunion by simulating the transport of a

tracer initialized in a minimal boundary layer approximation (PBL proxy) forward in time. This minimal PBL proxy was defined as an atmospheric layer located between the surface and 500 m a.g.l. but was capped at 1000 m above sea level (a.s.l.). Here, we use the reversed approach by considering backward trajectories and determining their presence in the PBL proxy. In order to resolve the capping at 1000 m a.s.l. in the FLEXPART-AROME output, a higher vertical resolution below 500 m a.g.l. was needed. By using the minimal PBL proxy we simulate the arrival of pollutants emitted into the boundary layer further away from the observatory. In order to quantify the impact of local emissions, a second PBL proxy which is not capped at 1000 m a.s.l. is used. Two approximations are applied here when using the SRR. Firstly, we accumulate the residence times over the complete period air parcels are transported over the specific source area for each release. By doing this, we do not consider a time dependence of emissions rates at the source. Since the largest temporal variation in emission rates manifests itself between day and night, coincident with the shift from PBL to FT air masses arriving at the observatory, we do not expect this approximation to significantly affect the results. Secondly, we separate the grid cells into three categorical variables (mountain, urban and marine) to identify the chemical signature of emissions at the source. The emission sensitivity to marine emissions is determined by air parcels located in the PBL proxy over the ocean. As urban areas are located near the coasts, generally with surface elevations below 500 m a.s.l., air parcels located in the PBL proxy over this region are categorized as sensitive to urban emissions. The remaining area is categorized as mountainous, with usually strong biogenic emissions during the day. Using the minimal PBL proxy by Lesouëf et al. (2011), the emission sensitivity in the mountainous area is restricted to a small band of surface elevation between 500 and 950 m a.s.l. By changing to the constant 500 m a.g.l. layer as a PBL proxy, we will only change the SRR of mountainous areas. We will discuss both the capped – PBL proxy by Lesouëf et al. (2011) – and uncapped – constant 500 m a.g.l. – mountain categories. In the following, discussion of the mountain category refers to the uncapped PBL proxy unless otherwise specified. By using the categorical variables, we neglect possible hotspots of anthropogenic sources related to industry or large cities. The relatively coarse resolution of AROME ($2.5 \times 2.5$ km$^2$) does not permit us to resolve local transport features induced by the strong orographic profile of La Réunion for individual measurements. As such, we will be using the model output to discuss median diel profiles and combine releases in the model to identify the impact of mesoscale transport on measurements at the Maïdo observatory.

## 3 Results and discussion

Due to the high altitude of the Maïdo observatory, it is primarily located in the free troposphere during the night. During the day, the pristine marine boundary layer air over the ocean is thermally driven towards the observatory by the sea breeze coupled to upslope transport on the western side of the island. This mesoscale transport is in competition with overflowing trade winds coming from the east. During the day, marine air masses pass over both anthropogenic and biogenic sources affecting the atmospheric composition. The dataset is split into nighttime – 22:00 until 05:00 LT – and daytime – 10:00 until 17:00 LT. As seen in Table 2, (O)VOC concentrations are generally higher during the day, except CH$_3$CN and DMS which present similar mixing ratios between nighttime and daytime. We will first describe and discuss the diel, seasonal, and inter-annual variability of tracers recorded with the hs-PTR-MS instrument (Sect. 3.1). Next, the PMF algorithm is used to attribute the atmospheric (O)VOC burden to specific sources (Sect. 3.2), which are identified by comparing temporal variations in the PMF source factor contributions to diel and inter-annual variability of the tracers reported in Sect. 3.1. Finally, the results obtained in Sect. 3.1 and 3.2 are compared to back-trajectory calculations performed with FLEXPART-AROME (Sect. 3.3) to further validate the PMF results and assess the impact of mesoscale transport on the atmospheric composition recorded at Maïdo.

### 3.1 Diel, seasonal, and inter-annual variability

Discussion on annual and diel variability is limited to a subset of VOCs – CH$_3$CN, C$_5$H$_8$, Iox, C$_6$H$_6$, C$_8$H$_{10}$, and DMS – of which the patterns can be unequivocally attributed to different sources – pyrogenic, primary biogenic, secondary biogenic, anthropogenic, and marine, respectively. To support this discussion we use Figs. 2–7 where the daily average concentrations, seasonal median diel profiles, and wind-separated (easterly versus westerly winds) median diel profiles for the different species under consideration and several meteorological parameters are shown.

### 3.1.1 Acetonitrile ($m/z$ 42, CH$_3$CN)

Biomass burning is the main source of CH$_3$CN in the atmosphere (de Gouw, 2003) and is often used as an indicator for BB plumes in atmospheric studies (e.g., Verreyken et al., 2020). Its sinks are reaction with the hydroxyl radical (OH) and uptake by the ocean surface. Both processes are slow, resulting in a long average atmospheric lifetime of CH$_3$CN, 1.4 years (de Gouw, 2003). This not only allows transport from distant sources (e.g., BB events in Africa, Madagascar, and also South America and Malaysia Duflot et al., 2010; Verreyken et al., 2020) but also allows the compound to be well-mixed in the atmosphere. From Fig. 2 we see that high concentrations of CH$_3$CN occurred during August in both

**Table 2.** Interquartile range of 30 min average (O)VOC mixing ratios (MR) recorded at the remote high-altitude Maïdo observatory during nighttime (22:00–05:00 LT, FT) and daytime (10:00–17:00 LT, remote PBL). The last column indicates the median 30 min average mixing ratio during nighttime with the standard deviation between parentheses. If the limit is below the limit of detection (LoD), we put < LoD.

| Compound | Daytime MR [pptv] | Nighttime MR [pptv] | Nighttime median MR ($\sigma$) [pptv] |
|---|---|---|---|
| $CH_3OH$ | 990–1531 | 525–887 | 724 (228) |
| $CH_3CN$ | 84–109 | 79–110 | 94 (39) |
| $CH_3CHO$ | 179–311 | 61–101 | 78 (41) |
| HCOOH | 557–1045 | 172–335 | 225 (474) |
| $CH_3COCH_3$ | 355–526 | 259–379 | 304 (152) |
| $CH_3COOH$ | 248–511 | 64–164 | 99 (226) |
| DMS | 11–20 | 7–16 | 11 (7) |
| $C_5H_8$ | 80–223 | < 9 | < 9 |
| Iox | 48–136 | < 3–8 | 4 (11) |
| MEK | 35–69 | 11–21 | 15 (10) |
| $C_6H_6$ | 12–25 | 4–11 | 6 (9) |
| $C_8H_{10}$ | < 10–21 | < 10 | < 10 |
| $C_{10}H_{16}$ | < 16–24 | < 16 | < 16 |

2018 and 2019. This is due to direct transport of African BB plumes towards the Maïdo observatory (Verreyken et al., 2020). After these first intrusions, the daily average mixing ratio is reduced to pre-BB plume intrusions (80–100 pptv). However, from September to November these daily averages are elevated (100–150 pptv) with spurious periods of high concentrations when BB plumes reach the observatory (up to 300 pptv). This period corresponds with the BB season identified from Fourier transform infrared (FTIR) measurements at La Réunion (Vigouroux et al., 2012). The elevated daily averages drop to a minimum from March to August (50 pptv) after which the pattern repeats itself. The seasonal diel profiles (Fig. 3) show no clear pattern except for the September–October–November season (SON) of 2017 and 2019 where there is a nocturnal (nighttime maximum) signature. If we assume that $CH_3CN$ is relatively well-mixed throughout the troposphere, this would suggest the existence of a marine sink. This was proposed from $CH_3CN$ measurements at the Mauna Loa observatory in Hawaii (Karl et al., 2003). However, a study of BB plume transport of African and Madagascan pyrogenic emissions towards the Maïdo observatory found that these plumes are primarily transported through the FT (Verreyken et al., 2020). The median nocturnal pattern during the SON 2017 and 2019 seasons could likely be due to the high number of BB plumes sampled during these periods. After the BB season (SON), the median diel profile is flat, and its mixing ratio steadily drops from December–January–February (DJF, 100 pptv) to June–July–August (JJA, 80 pptv). The wind-separated profile of $CH_3CN$ does not point to a difference in sources between the east or west (Fig. 4).

### 3.1.2 Isoprene ($m/z$ 69, $C_5H_8$)

Terrestrial biogenic emission of $C_5H_8$ strongly correlates to both temperature and solar radiation (Guenther et al., 1993). Due to its short atmospheric lifetime, of the order of 1 h (assuming an OH concentration of $2 \times 10^6$ molec. cm$^{-3}$), $C_5H_8$ mixing ratios at the observatory are largely due to emissions close to the observatory. The seasonal evolution of $C_5H_8$ displays low values in June–September and a gradual increase in the last months of the year (Fig. 2). This relates closely to the annual temperature variation as measured at the Maïdo observatory (Fig. 5). A clear diurnal (daytime maximum) pattern with mixing ratios increasing directly after sunrise (Fig. 3) is observed. The 2018 DJF season has lower $C_5H_8$ mixing ratios compared to the 2019 season. This pattern is also recorded for both the total solar radiation and the temperature during the same periods (Fig. 6). The DJF 2018 and 2019 diel profiles (Fig. 3) clearly illustrate the impact of the diel solar radiation and temperature profiles (Fig. 6). Due to the much drier and less cloudy 2019, the DJF season profiles of the relevant meteorological parameters and $C_5H_8$ mixing ratios are much more symmetrical than DJF 2018. The 2019 diurnal maximum during the SON season does not correspond with a maximum in temperature compared to other years (Figs. 3 and 6), but it does relate to a maximum in measured total solar radiation. During the FARCE campaign (April 2015), it was found that mesoscale transport had a large role in the $C_5H_8$ mixing ratios recorded at the Maïdo observatory (Duflot et al., 2019). Easterly winds transported air masses originating mostly from 2000 m a.s.l., thereby reducing the impact of surface emissions. Westerly winds are the result of thermally driven mesoscale transport and carry marine boundary layer (MBL) air masses over anthropogenic and biogenic sources towards the observatory.

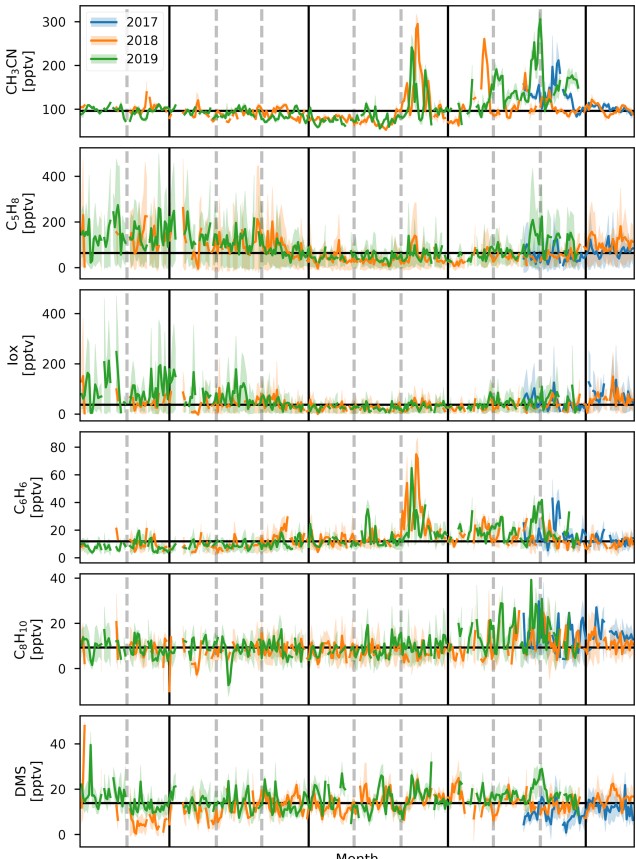

**Figure 2.** Daily average values of $CH_3CN$, $C_5H_8$, Iox, $C_6H_6$, $C_8H_{10}$, and DMS (top to bottom) for 2017, 2018, and 2019 (blue, orange, and green, respectively) during the deployment of the hs-PTR-MS instrument for the OCTAVE project. Shaded areas behind the curve show the diel interquartile range. Horizontal black line corresponds to the median daily average over the complete measurement period. Vertical gray dashed lines indicate the end of a month, and vertical black lines separate the DJF, MAM, JJA, and SON seasons.

As such, westerly (easterly) winds generally coincided with higher (lower) $C_5H_8$ mixing ratios, respectively. The difference in $C_5H_8$ concentration between the easterly and westerly flows was simulated with the non-hydrostatic mesoscale atmospheric model (Meso-NH) (Lac et al., 2018) during the FARCE campaign and was estimated to be $\sim 100$ pptv (Duflot et al., 2019). When separating the $C_5H_8$ mixing ratios over the entire dataset of this study according to wind direction, we found that the median diurnal profile is indeed slightly more elevated when the wind is coming from the west compared to the east (Fig. 4). The maximum difference in median diel profile between easterly and westerly regimes over the entire campaign is found to be 50 pptv. This takes into account the cold and dry season measurements where $C_5H_8$ mixing ratios are significantly lower than in April. Restricting our measurements to April 2018 and April 2019 (not shown) increases the discrepancy between wind regimes

to a difference of 150 pptv, with the highest mixing ratios recorded from westerly transport. This is most likely related to the closer proximity of Maïdo to vegetation west of the observatory.

### 3.1.3 Isoprene oxidation products ($m/z$ 71, $C_4H_6O$)

As Iox generally have a longer lifetime than their precursor, their concentrations are a measure for the impact of biogenic sources further away from the observatory. The daily average variation in Iox (Fig. 2) shows a similar signature as $C_5H_8$ with high daily average mixing ratios recorded in the DJF and March–April–May (MAM, up to 250 pptv) seasons. The median diel profiles (Fig. 3) show that the rise of Iox mixing ratios in the morning is delayed by about 1.5 h compared to $C_5H_8$, a duration similar to the expected isoprene lifetime, when assuming an OH concentration of $2 \times 10^6$ molec. cm$^{-3}$. Contrary to $C_5H_8$, Iox are elevated in the easterly wind regime (Fig. 4). This contradicts results from the FARCE campaign which suggested that air masses transported by easterlies originate mostly from 2000 m a.s.l. with only limited impact of surface emissions (Duflot et al., 2019) as the discrepancy in Iox is generally larger than that of $C_5H_8$. The stronger Iox in easterly winds compared to the westerlies could be related to either a relatively stronger sink of $C_5H_8$ from easterly transport – either a more distant source of $C_5H_8$ or elevated OH reactivity – or a larger source of $C_5H_8$ located to the east (either denser vegetation or larger area).

The median diel profile of the ratio between Iox and $C_5H_8$ separated according to the wind direction observed locally at the Maïdo observatory is shown in Fig. 8. As the mixing ratios of $C_5H_8$ are mostly below the limit of detection during the night, the nighttime baseline of about 0.5 of the Iox / $C_5H_8$ ratio roughly corresponds to the ratio of the instantaneous limit of detection of $C_5H_8$ (humidity dependent) to the mixing ratio of Iox (which is close to the detection limit) at these times (Table 2). After sunrise, when $C_5H_8$ mixing ratios sharply increase, the median Iox / $C_5H_8$ ratio decreases towards a minimum of 0.14. Afterwards, when air masses influenced by $C_5H_8$ emissions further away from the observatory are sampled, the ratio grows due to the increased oxidation during transport. At about 06:00 UT (10:00 LT) we note a difference between air masses originating east vs. west of the observatory. The ratio reaches a plateau at 09:00 UT (13:00 LT) of 0.50 or 1.00 when air masses originate west or east of Maïdo, respectively. After sunset, there is a sharp peak in the Iox / $C_5H_8$ ratio when air masses originate west of the observatory. This peak is not present when air masses come from the east. During the night, the planetary boundary layer is flushed away from the island, and the Iox / $C_5H_8$ ratio decreases until it reaches the value of about 0.50. This discrepancy between easterly and westerly winds may be partly explained by the meteorological conditions. Figure 7 shows that both total solar radiation and direct radiation are elevated

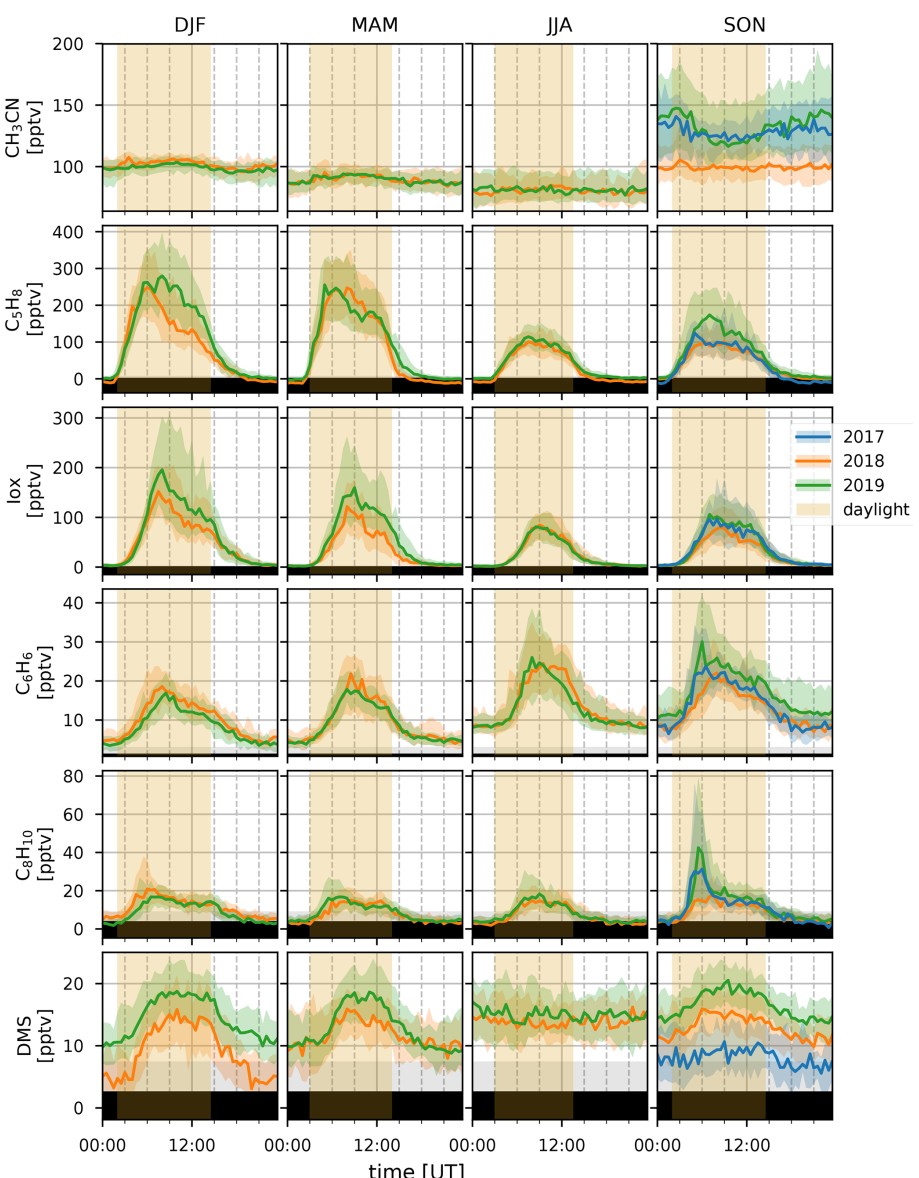

**Figure 3.** Median diel profiles (lines) and the interquartile distance (shaded area) for the DJF, MAM, JJA, and SON seasons (columns, left to right) in 2017, 2018, and 2019 (blue, orange, and green curves, respectively) for $CH_3CN$, $C_5H_8$, Iox, $C_6H_6$, $C_8H_{10}$, and DMS (rows, top to bottom). The yellow shaded area illustrates daylight during the respective seasons. Gray background shows the range between the median limit of detection and the minimum limit of detection of half-hour measurements. Individual half-hour measurements in the black region of the plots are not quantifiable.

in the afternoon when winds come from the east. This is due to reduced orographic cloud formation from the east which can enhance OH formation through photolytic reactions and therefore accelerate the $C_5H_8$ sinks during easterly transport. It is however unlikely that increased production of Iox is the only possible source of the discrepancy as a similar difference in radiation is found between the MAM seasons from 2018 and 2019 (Fig. 6). During MAM 2018, radiation at the observatory in the afternoon was slightly elevated compared to 2019, which could have enhanced the $C_5H_8$ sink and possibly Iox formation during MAM 2018 compared to

2019 (Fig. 3). However, Iox signals during MAM 2019 were increased compared to 2018. From Meso-NH simulations performed in the context of the FARCE campaign, it was found that $C_5H_8$ concentrations in the morning (06:00 UT, 10:00 LT) are highest on the northeastern part of the island (Duflot et al., 2019). The passage over this region of air masses arriving at the observatory from the east could explain the discrepancy between wind regimes.

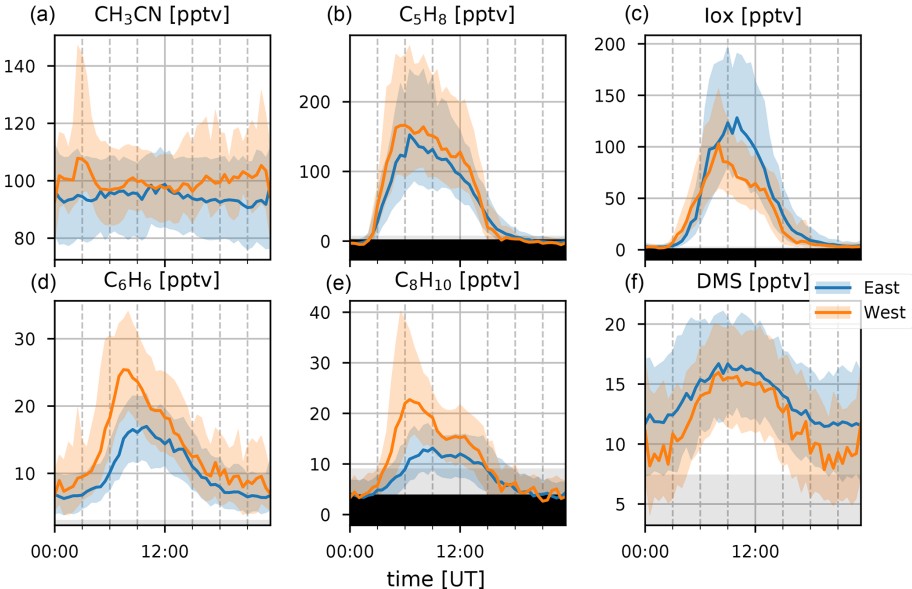

**Figure 4.** Median diel profiles (lines) and the interquartile distance (shaded area) for $CH_3CN$, $C_5H_8$, Iox, $C_6H_6$, $C_8H_{10}$, and DMS **(a–f)** separated between easterly (blue) and westerly (orange) transport recorded at the observatory. Gray background shows the range between the median limit of detection and the minimum limit of detection of half-hour measurements. Individual half-hour measurements in the black region of the plots are not quantifiable.

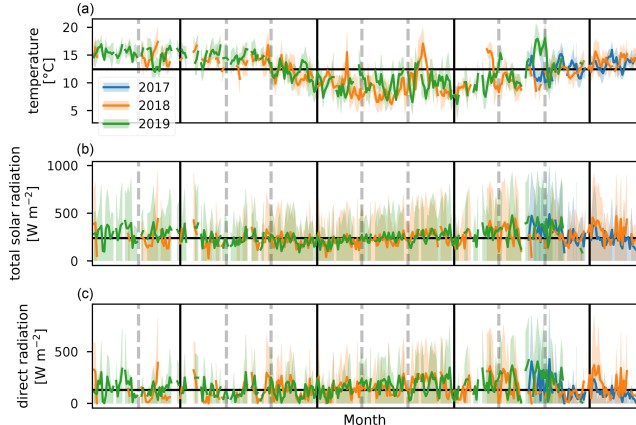

**Figure 5.** Daily average values of temperature, total solar radiation, and direct radiation **(a–c)** for 2017, 2018, and 2019 (blue, orange, and green, respectively). Shaded areas behind the curve show the diel interquartile range. The horizontal black line corresponds to the median daily average over the complete measurement period. Vertical gray dashed lines indicate the end of a month, and vertical black lines separate the DJF, MAM, JJA, and SON seasons.

### 3.1.4 Benzene and $C_8$-aromatic compounds ($m/z$ 79 and 107, $C_6H_6$, and $C_8H_{10}$, respectively)

The seasonal variation in $C_6H_6$ is linked to BB (Fig. 2). The excess $C_6H_6$ mixing ratios due to arrival of BB plumes in August 2018 and August 2019 have been studied in previous work (Verreyken et al., 2020). For $C_8H_{10}$, whose sources

are mostly fugitive emissions from industry, car exhaust, and volatilization through solvent use, this BB variability is not visible. Not only are emission factors of $C_8H_{10}$ at least 3 times smaller than those of $C_6H_6$ (Andreae, 2019), but the atmospheric lifetime of $C_8H_{10}$ (hours) is also much lower than that of $C_6H_6$ (weeks). $C_8H_{10}$ is strongly enhanced during the SON seasons of 2017 and 2019. The median diel profiles of this season indicate that the SON mixing ratios of $C_8H_{10}$ are enhanced in the morning (06:00 UT, 10:00 LT). This is most pronounced during the 2017 (maximum $C_8H_{10}$ mixing ratio of 30 pptv) and 2019 (40 pptv) seasons. The median diel profile of $C_6H_6$ does show a slight elevation of mixing ratios at the same time as for $C_8H_{10}$, although it is less evident given the large amplitude of the regular diel profile seen in the other seasons. Comparison of the $C_6H_6$ and $C_8H_{10}$ median diel profiles in westerly winds shows that the maximum in $C_6H_6$ occurs 1 h after the peak in $C_8H_{10}$ (Fig. 4). The $C_6H_6$ median diel profile corresponds best with the diel profile of emissions related to energy production at La Réunion in the EDGAR database (Crippa et al., 2020), which has a large peak in the morning (05:00 UT, 09:00 LT) and slowly decreases towards the evening (16:00 UT, 20:00 LT) when it drops quickly. This is in contrast to the median diel profile of $C_8H_{10}$, which corresponds best with the two-peak profile of emissions from both residential and road traffic sources (Crippa et al., 2020).

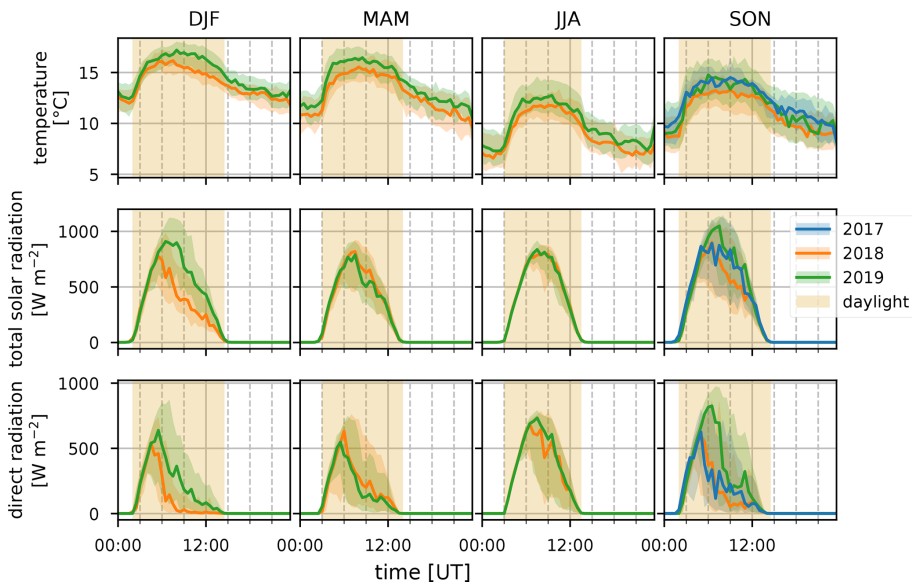

**Figure 6.** Median diel profiles (lines) and the interquartile distance (shaded area) for the DJF, MAM, JJA, and SON seasons (columns, left to right) in 2017, 2018, and 2019 (blue, orange, and green curves, respectively) for temperature, total solar radiation, and direct radiation (rows, top to bottom). The yellow shaded area illustrates daylight during the respective seasons.

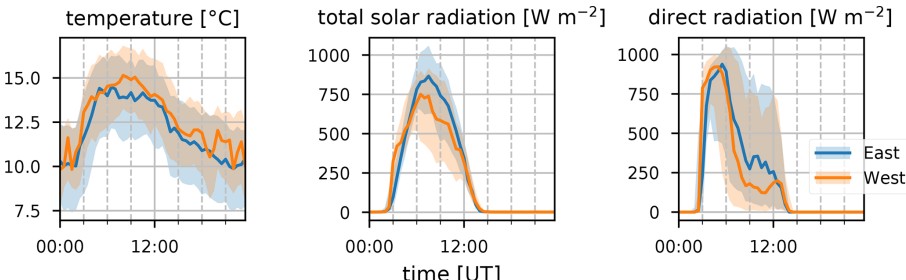

**Figure 7.** Median diel profiles (lines) and the interquartile distance (shaded area) for temperature, total solar radiation, and direct radiation (left to right) separated between easterly (blue) and westerly (orange) transport recorded at the observatory.

### 3.1.5 Dimethyl sulfide ($m/z$ 63, $CH_3SCH_3$)

DMS is predominantly produced by marine phytoplankton and emitted in the atmosphere from the ocean surface. High regional abundances of DMS are linked to the presence of phytoplankton blooms in upwelling waters (e.g, Colomb et al., 2009). It has been reported that trees may also be a net source of DMS in the atmosphere (Jardine et al., 2015; Vettikkat et al., 2020). Globally, these terrestrial emissions are negligible (370–550 Mg DMS yr$^{-1}$; Vettikkat et al., 2020) compared to the estimated annual marine emissions (28 Tg S yr$^{-1}$, which corresponds to 54.5 Tg DMS yr$^{-1}$; Lana et al., 2011). The main atmospheric sinks of DMS are daytime oxidation by OH, accounting for 73 %–84 % of the total sink and nighttime oxidation by the nitrate radical $NO_3$ (Berglen, 2004; Kloster et al., 2006). The global average atmospheric lifetime of DMS is 1.02–1.93 d (Berglen, 2004; Kloster et al., 2006). As La Réunion is lo-

cated in the tropics, the lifetime of DMS is expected to be shorter than the global average during the day (Blake et al., 1999). The DMS mixing ratios recorded at the Maïdo observatory are much lower (7–16 pptv in the FT during the night and 11–20 pptv in the planetary boundary layer during the day, Table 2) compared to reported measurements over the ocean (e.g., $60 \pm 20$ pptv from ship-borne measurements south of La Réunion, 24–30.2° S, during the MANCHOT campaign in 2004). The lower mixing ratios are likely due to the near absence of DMS sources over land in combination with oxidation by OH during daytime, when marine air can reach the observatory. At first sight, no clear seasonal pattern is visible (Fig. 2). However, the diurnal maximum in DMS diel profiles during the SON, DJF, and MAM seasons is not present during JJA (Fig. 3). The nighttime values vary between years for the SON and DJF seasons, for reasons still unclear. The diel profiles separated according to the wind direction (Fig. 4) suggest that westerly transport during the

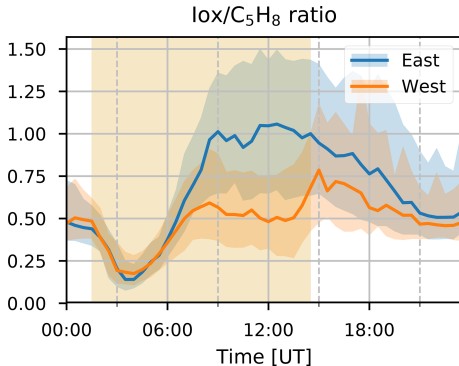

**Figure 8.** Median diel profile of ratio between isoprene oxidation products (Iox) and isoprene ($C_5H_8$), separated according to the wind direction observed at the Maïdo observatory: blue and orange for winds coming from the east and west, respectively. Interquartile ranges for each direction are shown as shaded areas in the background with colors corresponding to the curve. The shaded yellow area in the back corresponds to median daylight period over the 2-year observation period.

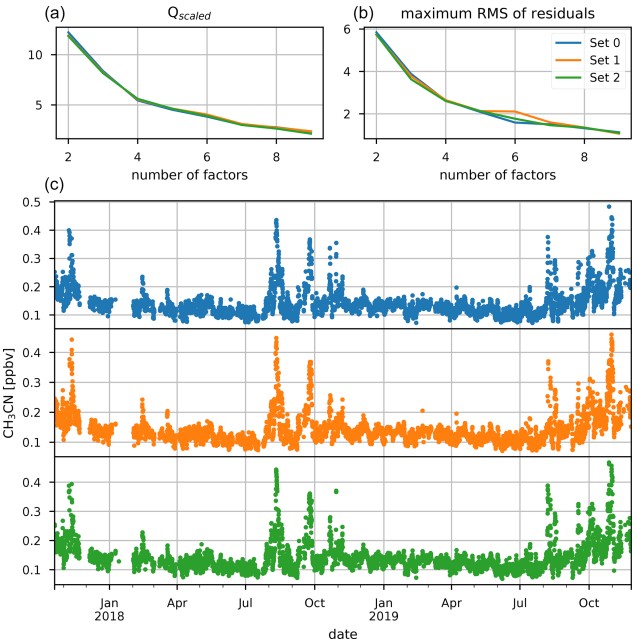

**Figure 9.** Dataset separation in three different subsets. The top two plots show the scaled objective function **(a)** and the maximal root mean square (rms) of compound residuals **(b)** for a scan of the number of factor space from 2 to 9. The bottom three plots **(c)** show the $CH_3CN$ mixing ratios (ppbv) for the three different subsets which ensure near-equal occurrence of biomass burning intrusions.

night is accompanied by somewhat lower mixing ratios (5–15 pptv) compared to easterly transport (8–16 pptv). Caution is required, however, as only a few nighttime measurements were recorded with a westerly origin.

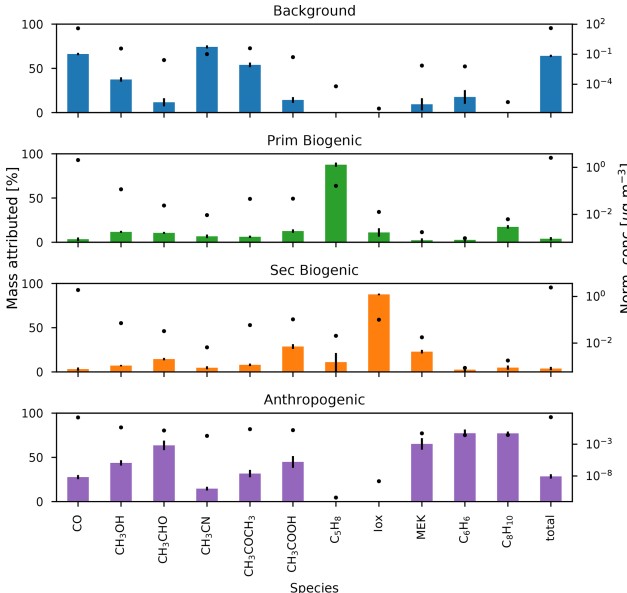

**Figure 10.** The median fraction of mass attributed (%) to the background, primary biogenic, secondary biogenic, and anthropogenic factors calculated by the bootstrapping algorithm (bars, interquartile distance shown by error bars) and the median normalized concentration ($\mu g\,m^{-3}$) of the factor profiles (markers).

## 3.2 Source identification using positive matrix factorization

The source apportionment software EPA PMF v5.0 was used to attribute the variability of VOCs to their sources. The PMF is run using CO data in combination with $CH_3OH$, $CH_3CN$, $CH_3CHO$, $CH_3COCH_3$, $CH_3COOH$, $C_5H_8$, Iox, MEK, $C_6H_6$, and $C_8H_{10}$ recorded with the hs-PTR-MS instrument. The 2.7 min time-resolved data from the hs-PTR-MS are accumulated over 1 h to match the temporal resolution of CO measurements and improve the signal-to-noise ratio ($S/N$) of the (O)VOC measurements. The mixing ratios are converted to mass concentrations [$\mu g\,m^{-3}$] for mass closure using local ambient temperature and pressure measurements. Data below the limit of detection (LoD) were set to $LoD/2$ with an associated uncertainty of $5/6 \times LoD$ in accordance with best practices (Norris et al., 2014). Measurements for which concentrations were missing for at least one of the species were taken out of the dataset instead of replacing the missing information by an average value with high uncertainty. Data quality is quantified by using $S/N$. Species with low $S/N$ ratio ($1.0 < S/N < 2.0$) were characterized as "weak" ($CH_3COOH$), and corresponding uncertainties were multiplied by a factor of 3 (Norris et al., 2014). Species for which data were not available for the entire campaign (HCOOH) were excluded from the analysis. Additionally, DMS was not used in the PMF analysis, as preliminary tests indicated that including DMS would result in unexplainable factor profiles and contributions. Uncertainties are cal-

culated using the displacement and bootstrapping algorithms. Analysis of the complete 2-year dataset by the EPA PMF 5.0 software was not possible as the dataset was too large. Instead, the dataset was randomly split up into three parts, while taking care to ensure equal contribution of BB intrusions in the three sets. This is checked using the temporal dataset of $CH_3CN$ concentrations (Fig. 9). Similar results for both the factor profiles and the contributions were obtained for all three subsets, and we report here only the results for one of the subsets. After scanning the parameter space with 25 base runs for each number of factors, we selected the four- or five-factor solution for further investigation based on the $Q_{scaled}$ and maximum rms of residual curves (Fig. 9). The base model was executed 100 times with a seed of 9 for both four and five factors. When using the displacement algorithm, no errors or swaps were reported using the four- and five-factor solutions, showing that they are free of rotational ambiguity. From the 100 bootstrapping runs for the four-factor solutions, it was found that 100 % of the BS factors were mapped to the base run. This is not the case for the five-factor solution (91 %), which implies that, although still a robust solution, it is slightly more sensitive to random errors. However, the five-factor solution had no straightforward interpretation and was therefore discarded. A further investigation of rotational ambiguity of the four-factor solution was performed using the $F_{peak}$ functionality. This is used to investigate the effect of flattening (positive $F_{peak}$ strengths) or sharpening (negative $F_{peak}$ strengths) the factor contributions. No decrease in the objective function was found using $F_{peak}$ strengths of ±0.1, ±0.2, ±0.5, ±1.0, and ±2.0, which confirms that the four-factor solution is free of rotational ambiguity. The four factors were identified as a background (which includes BB signatures), an anthropogenic, a primary biogenic, and a secondary biogenic factor. No marine source was identified from the PMF algorithm in part due to the omission of DMS data from the analysis.

### 3.2.1 Background and biomass burning factor

The background and BB factor accounts for 65 %–68 % of CO and 73 %–75 % of $CH_3CN$ measured at the observatory (Fig. 10). This is a strong indication that the factor is indeed a combination of background and BB signals. The annual pattern of the normalized contributions (Fig. 11) shows a strong influence of BB plumes reaching the observatory between August and November. No other seasonal pattern is present. The median diel profile of normalized contributions (Fig. 12) shows no influence of wind direction, which implies that the source is either near the observatory or remote. As the atmospheric lifetimes of both CO and $CH_3CN$ are long, the source is identified as remote, which is in agreement with the identification of the factor as background–BB. The nighttime maximum of the normalized factor contributions suggests that it originates in the FT. It may also indicate the presence of an ocean sink of $CH_3CN$, as proposed by Karl et al. (2003) and

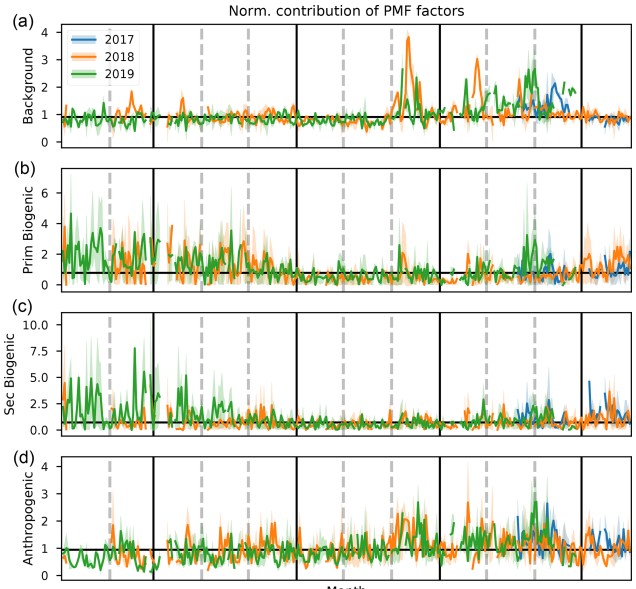

**Figure 11.** Daily average values of normalized contribution of the background, primary biogenic, secondary biogenic, and anthropogenic source factors **(a–e)** for 2017, 2018, and 2019 (blue, orange, and green, respectively). Shaded areas behind the curve show the diel interquartile range. The horizontal black line corresponds to the median daily average over the complete measurement period. Vertical gray dashed lines indicate the end of a month, and vertical black lines separate the DJF, MAM, JJA, and SON seasons.

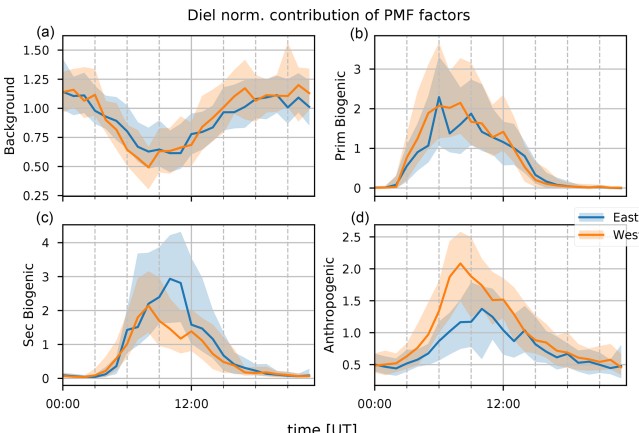

**Figure 12.** The median diel profile of background, primary biogenic, secondary biogenic, and anthropogenic factor **(a–d)** normalized contributions separated along the wind directions recorded at the observatory (blue for air masses coming from the east and orange from the west); hourly interquartile ranges are visualized by the shaded area behind each curve.

mentioned in Sect. 3.1.1, which is obscured by a compensating mesoscale source of $CH_3CN$, resulting in the recorded flat median diel profile. However, it seems unlikely that a mesoscale source would systematically compensate for this atmospheric sink perfectly. We think it is more likely that the

PMF algorithm artificially reproduces the flat diel profile of $CH_3CN$ (Fig. 3) when it is well-mixed between the PBL and FT by attributing a small fraction of $CH_3CN$ to other sources, inducing only small errors in the algorithm. The contribution of the background factor to the total mass of compounds included in PMF is 63 %–66 %. This is strongly biased by the high impact of CO concentrations that account for most of the mass of species included in PMF. Taking the contribution of CO out of the equation, we find that the background factor accounts for 33 % of the mass of (O)VOCs recorded at the observatory. The background factor is especially relevant for $CH_3OH$ (30 %–39 %) and $CH_3COCH_3$ (51 %–54 %). Note that, in general, the VOCs present in the background factor all have atmospheric lifetimes of at least several days. This factor does not contain compounds with short lifetimes (i.e., $C_5H_8$, Iox, and $C_8H_{10}$), which is expected as it originates mainly in the free troposphere and represents the impact of emissions from large bodies of land located far away from La Réunion.

### 3.2.2 Anthropogenic factor

The anthropogenic factor accounts for 25 %–29 %, 73 %–85 %, and 75 %–79 % of the mass of CO, $C_6H_6$, and $C_8H_{10}$, respectively (Fig. 10). This is a good indication that this factor is indeed related to anthropogenic sources. The strong daytime maximum indicates that this source is primarily related to anthropogenic activities located on the island. Thermally driven mesoscale transport features advect polluted air masses originating along the coastal regions towards the location of Maïdo during the day. The seasonal variation (Fig. 11) shows elevated normalized contributions in August–November. This may be due to either (i) excess $C_6H_6$ present in younger BB plumes not represented in the background–BB factor (especially in August) or (ii) the high $C_8H_{10}$ mixing ratios in SON (Fig. 2). The median diel pattern (Fig. 12) shows a large impact from westerly transport, with a peak between 06:00 and 09:00 UT (10:00–13:00 LT), similar to $C_6H_6$ (Fig. 4). This points to a strong influence of mesoscale transport on the contribution of an anthropogenic source to the local atmospheric composition recorded at Maïdo. The difference in diel profiles between $C_8H_{10}$ and $C_6H_6$ is not resolved by the PMF. Introduction of an additional factor did not result in a second anthropogenic source (e.g., by discriminating between combustion and evaporative sources as found at La Réunion during the OCTAVE intensive observation period (IOP); Rocco et al., 2020). The anthropogenic factor accounts for 38 % of the mass of (O)VOCs – i.e., excluding CO – recorded at the observatory. Besides $C_6H_6$ and $C_8H_{10}$ mentioned above, the anthropogenic source is dominant for $CH_3CHO$ (58 %–68 %), $CH_3OH$ (41 %–47 %), $CH_3COOH$ (38 %–51 %), and MEK (57 %–73 %) and is the second largest source for CO and $CH_3COCH_3$ (28 %–35 %) for which the background is the dominant contributor. As an independent verification,

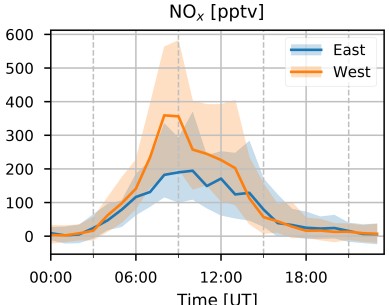

**Figure 13.** The median diel profile of hourly averaged $NO_x$ concentrations (pptv) along the wind directions according to measurements (blue for eastern transport and orange for western transport), recorded at the Maïdo observatory in 2018. Interquartile ranges are visualized by the shaded area behind the curves.

we note that the wind-separated diel profile of the anthropogenic source factor contributions strongly resembles the wind-separated median diel profile of $NO_x$ (Fig. 13). The $NO_x$ mixing ratios are much stronger when winds are coming from the west with a distinct peak just before 09:00 UT (13:00 LT). The difference in diel profiles between air masses advected along easterly–westerly flows is gone by 14:00 UT (18:00 LT). Both behaviors are also observed in the median diel profile of the normalized contributions for the anthropogenic source factor.

### 3.2.3 Primary biogenic factor

The primary biogenic factor accounts for 85 %–94 % of the mass of $C_5H_8$ recorded at the Maïdo observatory (Fig. 10). Its normalized contribution (Fig. 11) is enhanced during the hot and wet months (DJF and MAM). The median diel normalized contributions separated between wind regimes are found to be similar (Fig. 12). Together with the short atmospheric lifetime of $C_5H_8$, this implies that the primary biogenic factor is mostly determined by emissions close to the observatory. The primary biogenic factor accounts for 15 % of the mass of (O)VOCs recorded at the observatory. The primary biogenic source is the dominant source for $C_5H_8$ and the second largest source of $C_8H_{10}$ (15 %–19 %) and Iox (6 %–13 %). The first is unexpected as $C_8H_{10}$ is not usually associated with biogenic emissions. We must note, however, that the data quality of $C_8H_{10}$ is the lowest of all compounds included in the PMF algorithm. As the concentrations are low and the corresponding uncertainties are large, small discrepancies between the $C_8H_{10}$ variability and the contributions of the anthropogenic source factor are picked up by the other source factors with a daytime maximum. We do not expect this to be due to issues with colocation of biogenic and anthropogenic emissions or issues with separating different anthropogenic sources because (i) the consistency between air masses originating east and west of the observatory is consistent and indicates that this source factor is emitted locally

and not in coastal regions, and (ii) the biogenic source factor shows a large seasonal variability with strong contributions in austral spring and summer, which is not represented in the $C_8H_{10}$ profiles (Figs. 3 and 4). This indicates that the contribution of the primary biogenic source factor to the budget of $C_8H_{10}$ is most likely due to numerical uncertainties in the PMF algorithm which are not penalized sufficiently due to the relatively high uncertainties on their mixing ratios. The contribution of the primary biogenic factor towards the budget of Iox probably accounts for a fraction of $C_5H_8$ emitted from local sources that is rapidly oxidized. This would have a large impact, especially in the morning when $C_5H_8$ originating from emissions further away from the observatory cannot yet reach the observatory through mesoscale transport.

### 3.2.4 Secondary biogenic factor

The secondary biogenic factor accounts for 87 %–89 % of the mass of Iox (Fig. 10). The seasonal variation in the normalized contributions is similar to that of the primary biogenic factor (Fig. 11). The median diel normalized contribution profile however shows a significant impact of wind direction (Fig. 12). Similar to the diel profile of Iox (Fig. 4), the normalized factor contributions are elevated in the easterly wind regime. This shows, similar to the anthropogenic source, a significant impact of sources from the island (further away from the observatory) on the secondary biogenic factor. However, contrary to the anthropogenic source, a large source seems to be located east (not west) of the observatory. The strong seasonal dependence of contributions with the strongest contributions during austral spring and summer suggests that this factor is indeed related to biogenic activity located around the island. Furthermore, the strongest contributions originate east of the observatory, in strong contrast with the diel profiles of both $C_6H_6$ and $C_8H_{10}$ (typical anthropogenic tracers), which corroborates the identification of this source as a secondary biogenic source factor. The secondary biogenic factor accounts for 14 % of the mass of (O)VOCs recorded at the observatory. It is the second strongest contributor for $C_5H_8$ (0 %–14 %), MEK (21 %–26 %), $CH_3COOH$ (26 %–33 %), and $CH_3CHO$ (13 %–17 %). The contribution to $C_5H_8$ is expected to be due to residual mixing ratios lingering behind in the PBL before they are flushed from the surroundings of the Maïdo observatory by mesoscale transport features. The MEK mixing ratios at Maïdo are mostly attributed to the anthropogenic source factor; however, the secondary biogenic factor accounts for a large part of the total mass of MEK observed at Maïdo. This is consistent with observations of Yáñez-Serrano et al. (2016) where MEK mixing ratios were found to correlate well with both isoprene and its oxidation products at sites of a biogenic nature. This correlation was less pronounced at urban sites where MEK was predominantly related to anthropogenic activity (Yáñez-Serrano et al., 2016). As Maïdo is located in a large national park with anthropogenic activ-

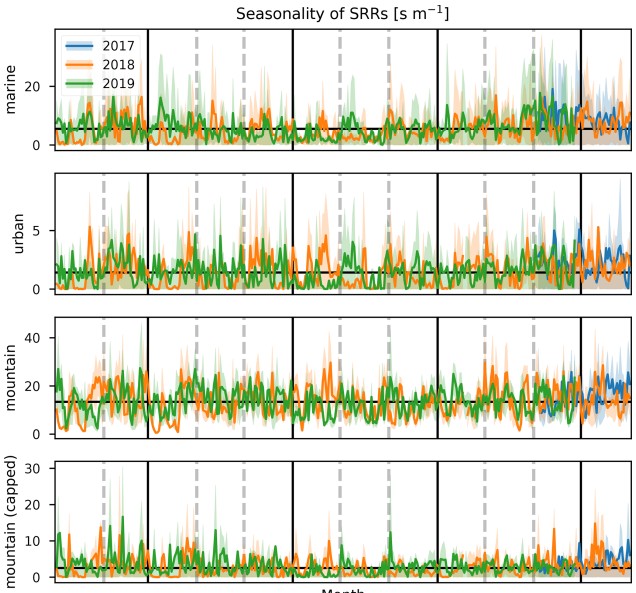

**Figure 14.** Daily average SRR $(s\,m^{-1})$ for marine, urban, mountain, and capped mountain categories (top to bottom) obtained from 12 h FLEXPART-AROME back-trajectory calculations for the years 2017 (blue), 2018 (orange), and 2019 (green). Interquartile distance is shown as the shaded area in the background.

ity located in coastal regions, it is expected that both the biogenic and anthropogenic sources may contribute to the total budget of MEK observed at this location.

### 3.3 Source identification using FLEXPART-AROME back-trajectory calculations

The FLEXPART-AROME back-trajectory calculations showed that air masses are equally sensitive to mesoscale sources located on the island during the first 12 h of the back-trajectory calculation compared to the complete 24 h. The SRRs for the 24 h trajectories had a significant impact only on the marine category. The average total residence time of air parcels in the first age class (12 h backward in time) was 11.6 h. This shows that a large fraction of air masses stayed within the domain during the first 12 h of simulation. The average total residence time of air parcels during the complete 24 h back-trajectory calculation was 17.7, illustrating that a large fraction of air parcels is transported outside the FLEXPART-AROME output domain. We will therefore limit our discussion to SRRs calculated during the first 12 h.

### 3.3.1 Diel, seasonal, and inter-annual variability

Figures 14 through 16 show the inter-annual daily averages, seasonal diel profiles, and wind-separated diel profiles of the categorical SRRs. The daily average emission sensitivities for the different categories (Fig. 14) do not show

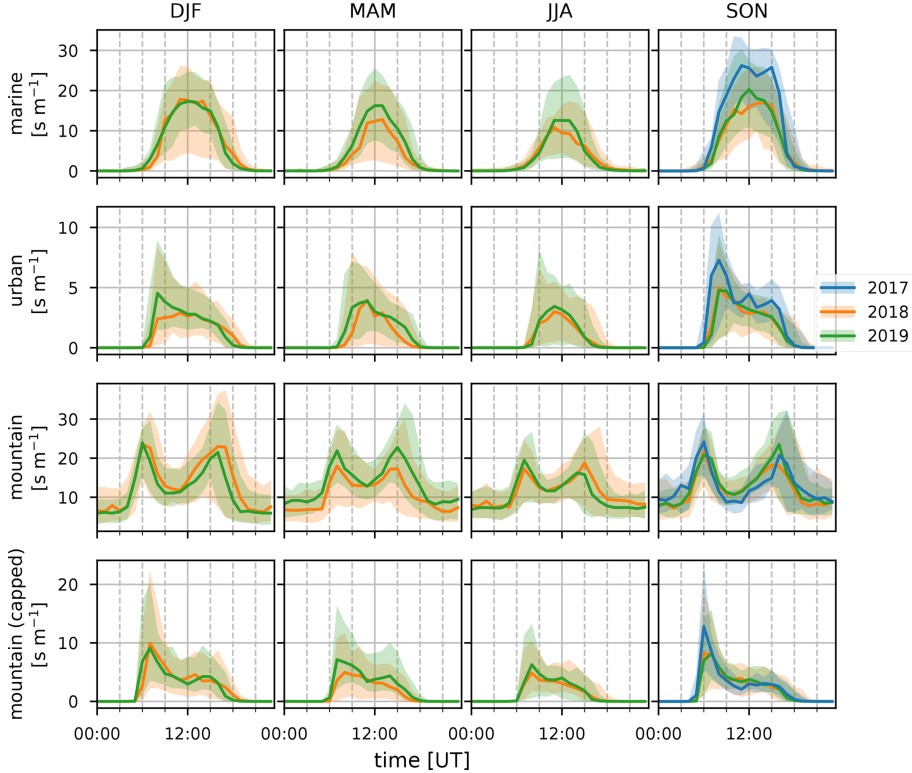

**Figure 15.** Median diel profile of the SRRs ($s\,m^{-1}$) calculated for the marine, urban, mountain, and capped mountain categorical emission sources (columns top to bottom) with FLEXPART-AROME 12 h back-trajectory calculations during the DJF, MAM, JJA, and SON seasons (columns left to right) in 2017 (blue), 2018 (orange), and 2019 (green). The interquartile distances are shown as shaded areas.

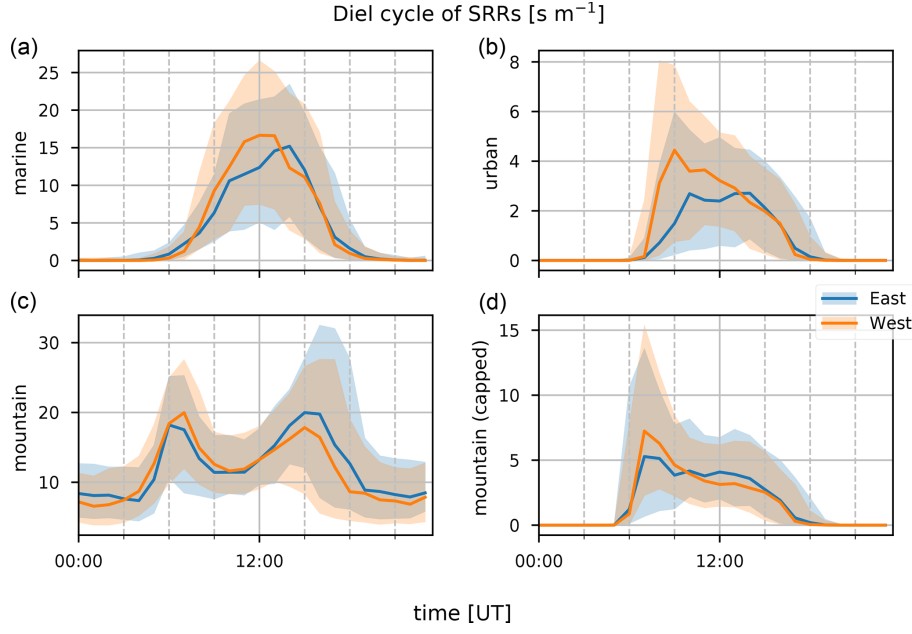

**Figure 16.** Wind-separated median diel profile (line) and interquartile distance (shaded area) for easterly (blue) and westerly (orange) transport for the marine **(a)**, urban **(b)**, mountain **(c)**, and capped mountain **(d)** emission source SRRs ($s\,m^{-1}$) as calculated using the FLEXPART-AROME 12 h back-trajectory calculations. Wind direction separation is based on measurements.

clear seasonal patterns for urban and both capped and uncapped mountain emissions. The sensitivity to marine emissions is lowest during the JJA season (Fig. 15). This is due to the strong trade winds during this time of year carrying air parcels outside the domain within the 12 h period. The monthly average total residence time in the first 12 h is second lowest of all (11.1 h). The month with the lowest average residence time (11.0 h) is January due to the impact of tropical storms and cyclones (see Table 1). There are however clear seasonal differences when considering the median diel profiles shown in Fig. 15. The urban SRR profiles generally have a morning peak located at 07:00 UT (11:00 LT). This peak is less pronounced during the MAM period and is not present during the JJA season. This may be due to the enhanced trade winds together with a weakening of the thermally driven mesoscale transport during austral winter, resulting in overflowing easterly transport affecting the observations at the location of Maïdo more frequently. This is also what is seen in the wind-separated diel profiles (Fig. 16) where the easterly regime has a low sensitivity to urban emissions before 12:00 UT (16:00 LT) compared to westerly transport. The diel profile for urban SRRs does not show the morning peak in the DJF season of 2018. The strong winds associated with tropical cyclones and storms may be the cause during this season. The wind-separated marine SRRs (Fig. 16) show stronger sensitivity from westerly transport between 07:00 UT and 13:00 UT (11:00–17:00 LT). This is counter-intuitive as we assume that overflowing easterly transport is organized along faster winds, and thus these air parcels spend less time over the island, which implies a relatively stronger impact of the ocean during the 12 h back-trajectories. However, the amount of land to cross for these air masses is much larger. Moreover, air masses originating west of the observatory travel along the coast of La Réunion and may have been trapped over the ocean in the wake of the island, thus increasing the impact of marine emissions before being pushed towards the observatory through the coupled sea breeze and upslope transport. The comparison between mountain emissions of the capped and uncapped categories manifests in two distinct features. Firstly, the median diel profile of the uncapped mountain SRR never reduces to zero at night. This implies that, according to the FLEXPART-AROME back-trajectories, the Maïdo observatory always has an impact of mesoscale PBL emissions and never measures purely free tropospheric air masses. The lack of a zero baseline is due to the location of the receptor site within the static PBL proxy for the uncapped representation in the model. The second and most striking difference is the presence of the double peak in the uncapped PBL representation compared to the single peak profile for the capped version. The first peak (present in both representations) is due to the reduced impact of the FT, which is coincident with the onset of upslope transport. Afterwards the PBL is diluted as the boundary layer top rises in altitude, thus decreasing the emission sensitivity to mountain sources (both capped and uncapped). After sunset, radiative forcing becomes zero, resulting in a compression of the PBL and a decrease in upslope transport. The reduced upslope transport results in a reduction of emission sensitivities linked to the capped mountain category. The uncapped mountain category on the other hand does not depend on this mesoscale transport and thus only shows the impact of the PBL compression which increases the SRR before the PBL is flushed by the FT. Note that the wind-separated diel profiles for mountain emissions are not dependent on wind direction (Fig. 16). This indicates that the discrepancy in the secondary biogenic source factor contribution between easterly and westerly flows (Fig. 12) is not related to transport of mountainous air masses and originates from (i) a stronger oxidative sink of primary biogenic emissions, (ii) air masses loaded with more biogenic emissions (denser vegetation/larger areas), or (iii) a combination of the above as discussed in Sect. 3.1.3. Here we can exclude the hypothesis of a larger source of biogenic emissions related to air masses passing over larger areas as this would be represented in the mountain SRRs.

### 3.3.2 Correspondence between FLEXPART-AROME and PMF

Table 3 shows the Pearson correlation coefficient ($r$) between the source–receptor relationship (SRR) of the different categories and the PMF source factor contributions. The correlations are generally low (maximum 0.45) due to the different approximations made in the categorization of SRRs and the neglected temporal variation in emission strengths, both diel and seasonal, at the source. The strongest correlation ($r = 0.45$) occurs between the urban SRR and the anthropogenic source contribution. This is due to the fact that the urban emissions have the most moderate seasonal profile. The low correlation is most likely due to the approximation of homogeneity of anthropogenic sources. The second highest correlation (0.40) is found between the secondary biogenic factor and the capped mountain source. It is expected that the capped mountain emission sensitivities correspond better with the biogenic (both primary and secondary) sources than the uncapped category. This is because of the dependence of biogenic emissions on solar radiation, which is not represented in the SRR. After sunset, the compression of the boundary layer results in a higher emission sensitivity from the uncapped mountain category. However, this increased sensitivity is compensated for by a rapid decrease in biogenic emissions after sunset, resulting in low mixing ratios of $C_5H_8$ (Fig. 3) and low primary biogenic factor contributions (Fig. 12). The better correlation of the SRR between the capped mountain category and the secondary biogenic factor contributions (0.40) compared to that with the primary biogenic factor contributions (0.25) is due to the occurrence of primary biogenic emission directly after dawn near the observatory. The oxidation of $C_5H_8$ to Iox takes time, similar to transport of air masses from the capped PBL mountain

**Table 3.** Pearson correlation coefficient ($r$) between categorical SRRs (m s$^{-1}$) and PMF normalized factor contributions.

| $r$ | Marine | Urban | Mountain (capped) | Mountain |
|---|---|---|---|---|
| Background | −0.08 | −0.19 | −0.25 | −0.18 |
| Anthropogenic | 0.38 | 0.45 | 0.35 | 0.20 |
| Primary biogenic | 0.20 | 0.23 | 0.25 | 0.11 |
| Secondary biogenic | 0.36 | 0.36 | 0.40 | 0.20 |

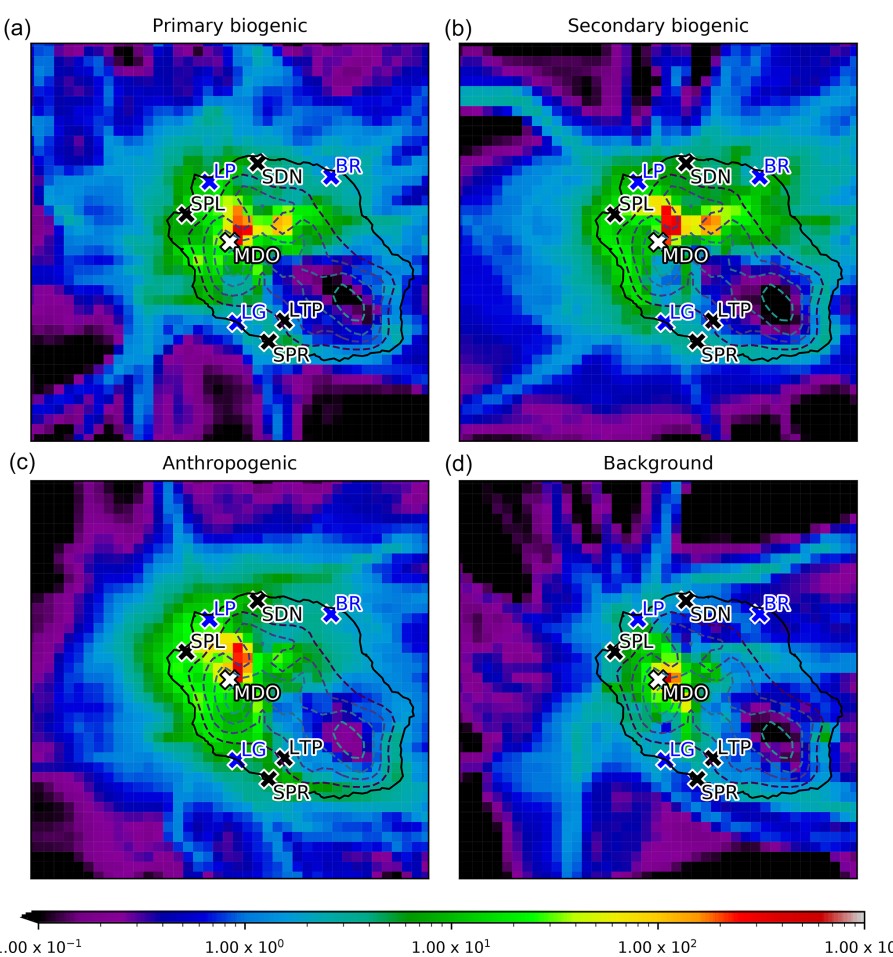

**Figure 17.** The spatial distribution of SRRs (s m$^{-1}$) over La Réunion for the top 5 % source factor contributions calculated by the PMF algorithm for the primary biogenic **(a)**, secondary biogenic **(b)**, anthropogenic **(c)**, and background **(d)** sources. The Maïdo observatory (MDO) is indicated in white. Cities with population of over 50 000 habitants – i.e., Saint-Denis (SDN), Saint-Paul (SPL), Le Tampon (LTP), and Saint-Pierre (SPR) – are shown in black. The largest industrial sites – Le Port (LP), Bois-Rouge (BR), and Le Gol (LG) – are indicated in blue. Dashed lines are surface elevation contours in the FLEXPART-AROME model separated by 500 m.

category towards the Maïdo observatory. Figure 17 shows the spatial distribution of SRRs (i.e., not summed over the spatial index), also referred to as the field of view (Seibert and Frank, 2004), using the uncapped PBL proxy accumulated over the largest 5 % source factor contributions from PMF source factors. We note that the total surface sensitiv-

ity is smallest for the background factor ($4.7 \times 10^3$ s m$^{-1}$); other total surface sensitivities are $7.2 \times 10^3$, $8.8 \times 10^3$, and $9.1 \times 10^3$ s m$^{-1}$ for the primary biogenic, anthropogenic, and secondary biogenic sources, respectively. As discussed in Sect. 3.2.1, the background factor contributions are a combination of air transported predominantly in the FT and BB

plume signals. As the BB plumes from Africa and Madagascar are mainly transported towards the Maïdo observatory through the FT (Verreyken et al., 2020), we expect the largest 5 % factor contributions to have a small SRR with mesoscale areas resolved from FLEXPART-AROME back-trajectories. From the similar median diel profiles of the wind-separated normalized factor contribution for the primary biogenic factor (Fig. 12), we know that emissions are mostly located close to the observatory. However, from the $C_5H_8$ wind-separated profile (Fig. 4), we see that there is a slight discrepancy with higher mixing ratios originating west of the observatory. This is contradicted somewhat by Fig. 17 with a maximum east of the observatory. This location corresponds well with the region of the highest $C_5H_8$ mixing ratios modeled with Meso-NH during the 2015 FARCE campaign (Duflot et al., 2019). This biogenic emission hotspot is also visible in the field of view corresponding to the strongest secondary biogenic factor contributions where a strong impact of surface emissions is seen at the same location (Fig. 17). A second feature in this map is the high impact north of the observatory which diverts to the west towards Le Port and is then transported south. This transport is organized along the canyon of the river "Rivière des Galets" between Mafate and Le Port (Fig. 1). The important role of river canyons with regard to pollutant transport towards the Maïdo observatory has already been identified in the forward transport study using Meso-NH (Lesouëf et al., 2011). The field of view of the observatory related to the strongest anthropogenic factor contributions is highest west of the Maïdo observatory. The observatory is thus most sensitive to emissions from the industrial centers at Le Gol and Le Port and the second largest city Saint-Paul. Other emission hotspots of human activity did not contribute as strongly to the top 5 % of anthropogenic source factor contributions.

## 4   Conclusions

In this study, we analyzed the 2-year dataset of (O)VOC concentrations obtained at Maïdo observatory between October 2017 and November 2019 in the framework of the OCTAVE project by a combination of (i) diel, seasonal, and inter-annual VOC concentration patterns; (ii) a PMF algorithm; and (iii) the FLEXPART-AROME mesoscale transport model. The measurements are shown to be useful to characterize the atmospheric background composition, as evidenced by the high average impact of the background–biomass burning source factor identified using the PMF algorithm on the atmospheric (O)VOC load (33 %), which is even larger during the night. During the day however, the atmosphere is loaded with anthropogenic and both primary and secondary biogenic tracers. The atmospheric loading by anthropogenic sources (38 %) is dominant compared to biogenic sources (29 %). This dominance is likely related to the short lifetime of $C_5H_8$ and its oxidation products. No marine

source was identified from the PMF algorithm for the burden of the VOCs considered here, in part due to the omission of DMS data from the analysis. At night, the contribution of mesoscale sources is strongly reduced, and the background factor becomes dominant. This diel variability is consistent with the description of mesoscale transport in previous studies (Lesouëf et al., 2011; Baray et al., 2013; Guilpart et al., 2017; Foucart et al., 2018; Duflot et al., 2019). The observatory is located near a horizontal wind shear front between overflowing trade winds coming from the east and the coupled thermally driven sea breeze with upslope transport during the day (Duflot et al., 2019). A previous study however found that overflowing trade winds were correlated with a reduced influence of surface emissions (Duflot et al., 2019). This is not reproduced here, as overflowing winds coming from the direction of the Maïdo mountain summit during the day were still sensitive to surface emissions. At night however, the cold island surface pulls down FT air masses towards the observatory, limiting the impact of mesoscale surface emissions on the atmospheric composition. This is consistent with known mesoscale transport features (Lesouëf et al., 2011; Baray et al., 2013; Guilpart et al., 2017; Duflot et al., 2019). The interquartile range of nighttime concentrations, characteristic for the free troposphere, of $CH_3OH$, $CH_3CN$, $CH_3CHO$, $HCOOH$, $CH_3COCH_3$, $CH_3COOH$, and MEK were found to be 525–887, 79–110, 61–101, 172–335, 259–379, 64–164, and 11–21 pptv, respectively. Back-trajectories initialized at the location of Maïdo, calculated with the mesoscale Lagrangian particle dispersion model FLEXPART-AROME, showed that the observatory is equally sensitive to island emissions 12 h backward in time or 24 h backward in time. By using a naive categorization of the island surface identifying coastal regions as areas with human activity, the model predicts a higher impact of anthropogenic emissions in air masses coming from the west compared to air masses advected towards the observatory from the east. This was in line with results from both the behavior of specific tracers recorded with the hs-PTR-MS instrument and the anthropogenic source factor from PMF which indicates that the diel discrepancy is related mostly to transport of coastal air masses towards the observatory. The sensitivity to generalized mountain regions, home to a multitude of different ecosystems over the island, did not reveal a higher sensitivity in air masses coming from the east, found from observations and PMF, which points to a source with enhanced biogenic emission rates east of the observatory. Combining results from FLEXPART-AROME and the PMF analysis, we identified emissions hotspots with significant impact on the atmospheric composition at the Maïdo observatory. Biogenic emission hotspots identified from both the highest contributions of primary and secondary biogenic source factors revealed a region east of the observatory, together with the "Rivière des Galets" river canyon, as important sources of biogenic tracers recorded at the observatory. The most relevant anthropogenic emission hotspots are located west of

the observatory. Specifically, industrial regions such as Le Port and Le Gol, together with the second largest city on the island, Saint-Paul, were found to have a large influence on the atmospheric composition at Maïdo. The background–biomass burning source factor from PMF was much less sensitive to surface emissions on the mesoscale compared to other sources.

*Data availability.* The core hs-PTR-MS dataset is available online (Amelynck et al., 2021, https://doi.org/10.18758/71021061). CO measurements are available from https://doi.org/10.18160/X22K-CP0G (De Mazière et al., 2020). NO$_x$ data for 2018 are currently under review for submission to the ACTRIS infrastructure but are available upon request from the PI (A.C.).

*Supplement.* The supplement related to this article is available online at: https://doi.org/10.5194/acp-21-1-2021-supplement.

*Author contributions.* The analysis was conceptualized by CA, JB, and BV. The formal analysis and visualization were performed by BV. Data acquisition, quality assurance, and processing of hs-PTR-MS data were done by CA, NS, and BV. The PMF analysis and FLEXPART-AROME simulations were performed by BV. Data acquisition, quality assurance, and processing of NO$_x$ data were done by JMM and AC. Data management of the meteorological parameters and the PICARRO instrument was carried out by CH and NK. Technical support at the Maïdo observatory for instrumentation for meteorological measurements and the PICARRO instrument was done by JMM. The original draft was prepared by BV in cooperation with CA, JFM, TS, JB, and NS. All co-authors were involved in the review and editing process of the final paper version.

*Competing interests.* The authors declare that they have no conflict of interest.

*Acknowledgements.* This research was funded by the Belgian Federal Science Policy Office (grant no. BR/175/A2/OCTAVE) and received extra financial support from Horizon 2020 (ACTRIS-2, grant no. 654109).

*Financial support.* This research has been supported by the Belgian Federal Science Policy Office (grant no. BR/175/A2/OCTAVE) and the Horizon 2020 (grant no. ACTRIS-2 (654109)).

*Review statement.* This paper was edited by John Liggio and reviewed by two anonymous referees.

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
