# Peer review of "Measurement report: Source apportionment of volatile organic compounds at the remote high-altitude Maïdo observatory"

_Atmospheric Chemistry and Physics, 2021_

## Author Comment (AC1)

We would like to thank the reviewers for their work on reviewing the manuscript. We have taken their comments into account when working on a new version of the manuscript. Below we respond to the specific remarks from the referee reports.

Before starting, we first identify some issues that we have noticed ourselves. First, the caption for Table 1 does not only include periods where the PTR-MS was shut down but rather shows events that might impact the atmospheric situation at Maïdo (meteorological or chemical composition). Second, in L119, it is stated that the inner diameter of the sampling line was 3/8 inch. This is the value for the outer diameter.

Both issues have been corrected in the revised version of the manuscript.

**Specific comments referee #1:**

*I would recommend that the authors avoid unnecessary abbreviations that are also not frequently used include FT, NWP, ACs, RTs, DISP, BS, AC, and more… I think using the original wording would improve a lot the readability of the paper since I found myself many times trying to find what the abbreviations meant.*

We have changed the less frequently used abbreviations to the original wordings. We did however opt to keep FT in contrast with PBL.

*Line 108: 81 is a fragment of monoterpenes while 137 is the original monoterpenes m/z. E/N is high enough for the instrument to possibly mostly see the signal in 81 but that might be something to mention here.*

Proton transfer from $H_3O^+$ to a monoterpene in the drift tube reaction of the PTR-MS indeed mainly results in both the protonated monoterpene at m/z 137 ($H^+.C_{10}H_{16}$) and a fragment ion species at m/z 81 ($C_6H_9^+$) and the contribution of the fragment will indeed increase with E/N. However, as the latter ion species may also be partially related to dissociative proton transfer to other compounds (e.g. hexenals or other $C_6$ compounds), we preferred to use the ion signal at m/z 137 for monoterpene quantification.

*Line 111-113: I am surprised to see 93 in there which I would expect to be most affected by the toluene signal which should not be long-lived and therefore not have high background. Any ideas why?*

With "background values", we were referring to the ion signals at m/z 93 during zero VOC measurements, i.e. the instrumental background. At low ambient concentrations of toluene (e.g. at night), ambient ion signals at m/z 93 were found to be significantly lower than the zero-VOC ion signals, resulting in strongly negative calculated concentrations. This suggests production of a compound in the catalytic converter or elsewhere in the setup and resulting in an ion signal at m/z 93. For this reason, this ion species has not been further considered in the analysis.

*Line 117-118: You are still expecting MVK and MACR. Since this instrument cannot separate the contribution of the different compounds, I would promote changing the naming to something like e.g., secondary oxidation products. This would be something to change throughout the paper.*

Although MVK+MACR concentrations have also been linked to primary emissions from automobile exhaust (e.g. Biesenthal et al., 1997) or biomass burning (e.g., Hatch et al., 2015), we expect that MVK and MACR in the present study are predominantly originating from the oxidation of isoprene near the location of Maïdo. This is also clear from the temporal variation of the m/z 71 ion signal which is reflecting a strong seasonality also observed for isoprene mixing ratios. As such, we prefer to keep

associating the term "isoprene oxidation products" to this ion signal. However, a more nuanced introduction to this terminology is given in the manuscript.

*Line 129: A graph in the SI to support this comparison would be nice to have.*

We have added information on the comparison between calibrations using both instrumental setups in supplementary material.

*Line 131: A sentence of two to briefly describe the indirect measurements would be informative here.*

More information has been added to the manuscript.

*Line 135: Discussion of the overall measurement uncertainties based on the calibrations and humidity correction would be nice to have in a table. For Iox I would expect the calibration factor for MVK compared to ISOPOOH to be drastically different. It would be nice to add some sentences regarding the uncertainty for accurate measurements for this m/z.*

We have added a paragraph to discuss the error analysis in the supplementary information.

As MVK and MACR are isomeric compounds, they cannot be separated by PTR-MS when using $H_3O^+$ precursor ions. However, since they both result in the protonated compound (100%), and since their proton transfer rate constants with $H_3O^+$ are very similar (3.8 and 3.6 $cm^3$ $molecule^{-1}$ $s^{-1}$, Zhao and Zhang, 2004), their detection sensitivity is equally similar. The sum of those two compounds can therefore be accurately quantified by calibrating the instrument against a known concentration of one, or both, of those compounds. In our setup, both MVK and MACR were present in the calibration bottle with equal mixing ratios of 500 ppbv in $N_2$.

Isoprene hydroxy hydroperoxide (ISOPOOH), another OH-initiated oxidation product of isoprene which is mainly relevant at low $NO_x$ conditions, is (partly) converted into either MVK or MACR (ISOPOOH isomer dependent) by interacting with surfaces inside the PTR-MS instrument. Consequently, ISOPOOH also contributes partially to the ion signal at m/z 71. Without further knowledge on local ISOPOOH concentrations, it is not possible to isolate the contribution of ISOPOOH to the ion signal at m/z 71. In order to take this potential interference into account, we opted to refer to the compounds resulting in m/z 71 ions as isoprene oxidation products (Iox) and inferred their mixing ratio by taking into account the calibration factor for MVK+MACR.

That being said, based on measurements performed during the SEAC[4]RS campaign (details on this campaign in Toon et al, 2016), the ratio ISOPOOH/Iox can be roughly estimated at Maïdo. SEAC[4]RS included measurements of both Iox (from a PTR-MS instrument) and ISOPOOH (from another instrument). Based on their results, we can retrieve the ISOPOOH/Iox ratio as a function of the NOx concentration. By assuming a similar relationship at Reunion Island, and using the NOx concentration measured at Maïdo, we find that ISOPOOH/Iox could be as large as 30% in the morning, the evening, and during the day when winds are coming from the East. At this time, NOx mixing ratios at the observatory are about 100-200 pptv. When the air masses originate from the West, NOx mixing ratios are 200-600 pptv, which corresponds with ISOPOOH/Iox ratios of 30-14% (median of 350 pptv NOx corresponds to a ratio of about 20%).

By considering a detection efficiency of 44% for 1,2-ISOPOOH at m/z 71 compared to MVK (Rivera-Rios et al. 2014) and assuming a maximum ISOPOOH/Iox ratio of 30%, the interference of ISOPOOH could lead to a maximum overestimation of MVK/MACR of only 13%. However, measurements of the SEAC[4]RS were performed over the U.S.A. and the ISOPOOH/Iox versus NOx relationship may not directly be applicable at La Réunion. As such, we have opted not to include this information in the

manuscript, nor in the supplementary information but use this here to reply to the referee's comments.

*Section 2.3.2: I find this section hard to read and challenging to follow the details of the approach used in this study without reading the cited publications. I feel this could be improved if the authors elaborated a bit more on the model they use and assume that readers haven't read and don't have to read in detail the mentioned publications. Introduction to what each model does in detail and the benefits of combining the two models would be great. Characteristic examples that I found hard to follow were line 173-175, line 183-185, line 195-197.*

Efforts have been made to clarify this section in the manuscript.

*Line 193-194: Is this done assuming an emission rate? How is this derived?*

This method is used for inverse modelling studies in order to estimate the emission rates at the source. Here, however, we opted to restrict ourselves to the discussion of the source-receptor relationships (SRRs) and the discussion here is to aid the reader in interpreting the SRRs. We have clarified this in the manuscript.

*Line 232: Are the authors capable of proving this is only secondary biogenic and not in general secondary? If not, I would recommend broadening to just secondary.*

The seasonal behaviour of the mixing ratios reflects well the seasonality of isoprene, which is emitted predominantly from the biosphere. Additionally, separating the diel profiles according to the wind direction clearly illustrated that the diel pattern of Iox reveals a large source east of the observatory which is relatively depleted from anthropogenic tracers. Moreover, from the footprint of the strongest 5% contributions of the PMF factors related to biogenic activity, we see clear similarities in emission hotspots between the factors responsible for a majority of isoprene (the primary biogenic factor) and the majority of the isoprene oxidation products (secondary biogenic factor). These are clear indications that the signal at m/z 71 is indeed predominantly due to oxidation of compounds emitted by the biosphere.

*Section 3.1.1-3.2.5: In the section naming, I suggest providing the chemical formula and the possible compound name as a more precise representation of what can be measured with a PTR-MS. Also, ratios of individual compounds, especially of anthropogenic nature, to CO might be informative in identifying pollution sources. Did the authors check these ratios and compare them to other emission sources? Comparison to other studies and inventories to improve the discussion of each section would be a great addition here.*

We have adjusted the titles of the different sections in the manuscript to include the ion m/z ratio and the chemical formula of the compound(s) characterised. We checked out the ratios with CO, specifically for the anthropogenic tracers. However, as anthropogenic sources located on the island are concentrated near the coast, co-location of the anthropogenic emissions, as well as the influence of biomass burning, makes it difficult to unambiguously identify different patterns without identifying special case studies, which is outside of the scope of the current work. Instead, we opted to focus on source attribution using a PMF algorithm, which unfortunately was unable to resolve different signatures related to anthropogenic activities.

*Section 3.1.3: I would consider changing this name to secondary oxidation products and moving this section to 3.1.5 for the reader to have a better understanding of the primary emission trends first before*

*discussion secondary sources. Also, this section is focusing the discussion on secondary biogenic sources alone and completely misses all possible sources of MVK and MACR. Although biogenic oxidation may contribute to the signal, anthropogenic sources in the island could impact the observed signal and trends. For example, I find it interesting that the trends of Iox are matched with trends of C₈H₁₀. Further discussion on this would clarify more the impact of different pollution sources on these trends.*

As mentioned before, patterns emerging from Iox are best related to isoprene as evidenced by the seasonal patterns, wind-separated diel profiles, and the correspondence in footprints between the primary biogenic and secondary biogenic source factors. Correspondence between trends of Iox and $C_8H_{10}$ may be due to mesoscale transport features transporting air masses from coastal regions towards the observatory during the day. As a result, timing between oxidation of isoprene to Iox and the arrival of air masses originating from urban regions may correspond. This could explain some similar features between Iox and anthropogenic tracers. As such, we kept the current structure as it is most beneficial to discuss Iox after the discussion of the profiles of their precursor.

*Line 355-357: Do you mean that random hours of the day were chosen to reduce the dataset length of 3 different PMF inputs to reduce the length of the data to 1/3? If so I would recommend rephrasing especially since the reader cannot tell much by Figure 8. For Figure 8 I would also consider providing all timeseries together in one panel and zooming in to one specific plume (e.g., August-September-October) using a different subpanel to highlight the differences in the data chosen per run.*

The dataset was split in three random subsets. Specifically, we generated a random number 1, 2, or 3 for each data point to designate to which subset it was assigned. If we would show the data shown in the lower three panels of Figure 8 in one plot, no clear information is obtained either. The point of these panels is that, although every point in our dataset is allocated to only 1 subset, we are not able to make a clear distinction between the different subsets. As such, we expect the PMF to perform similarly between the three different subsets of data. To clarify that each point only belongs to 1 subset of data, we have added a plot to the supplementary material where we show the split on the same plot for eight days during August 2018.

*Line 387: Please elaborate a bit more on why these compounds are present in the BG factor. Is this something expected? I would think so but a discussion here would be great.*

The VOCs present in the background factor all have atmospheric lifetimes of at least several days. This factor does not contain compounds with short lifetimes, which is expected as it originates mainly in the free troposphere and represents the impact of emissions from large bodies of land located far away from La Réunion. This information has been added in the manuscript.

*Section 3.2.2: Do the authors expect urban and industrial emissions from human activity from the island itself to play no role in the observed trends? I think a discussion on the local vs. long-range emissions detection would be valuable here. For example, when the back trajectory analysis indicates sources originating from the island's industrial or urban areas, does the anthropogenic PMF factor increase? Graphs that highlight that would be great. Reading through the paper I see that Figure 15 already does that to some extent. Wouldn't this therefore further supports the influence of local sources?*

Indeed, we expect the anthropogenic factor to be mainly due to anthropogenic activities located on the island. Long-range interferences picked up by this factor are related to biomass burning plumes which are not completely represented by the background/biomass burning factor. The strong daytime maximum already hints to the fact that this source factor is due to mesoscale sources

located along the coast. This maximum is due to the mesoscale transport features which advect polluted air masses from the coastal region to the location of Maïdo during the day.

*Section 3.2.3: How confident are the authors that the anthropogenic and biogenic emissions are fully disentangled? Based on their diurnal patterns I would expect that the PMF has a hard time separating co-emitted sources that are both expected to increase midday. Could that be the reason for the increased contribution of $C_8H_{10}$ here?*

We must note here also that the data quality for $C_8H_{10}$ is the lowest of the compounds taken into account for the PMF. As the non-anthropogenic source factors only contain small concentrations of $C_8H_{10}$, these factors only have a small impact on the simulated profile. Consequently, $C_8H_{10}$ attribution suffers most from numerical fluctuations in the PMF analysis. These fluctuations are not sufficiently penalised due to the relatively large uncertainties of $C_8H_{10}$ mixing ratios thus resulting in contributions of unexpected source factors.

*Section 3.2.4: I think that this factor is not discussed enough and by the current discussion the naming should change to secondary oxidation rather than biogenic. Also, it would be great if the authors could discuss the contribution of other compounds in this factor including $CH_3CN$, $C_5H_8$, and MEK. Is this related to the challenges of the PMF separating different sources?*

As noted before, Iox is expected to be mostly due to oxidation of isoprene, emitted by the biosphere. As such, we will keep the naming as biogenic secondary source. Isoprene contributions in this factor are expected to account for residual isoprene concentrations in the evening lingering behind in the planetary boundary layer mixture before it is flushed from the observatory due to mesoscale transport. It has already been shown that observations under a strong influence of biogenic emissions may contain elevated MEK concentrations (e.g. Yáñez-Serrano, 2016), corresponding well with mixing ratios of both isoprene and its oxidation products. As such it is not unrealistic that a fraction of MEK can be attributed to this source. Lastly, contributions of the secondary biogenic source to $CH_3CN$ are very low and are expected to be due to a compensating effect to account for the flat diel profiles similar as the contributions of $CH_3CN$ in the other non-background/biomass burning factors.

*Section 3.3: I had a hard time following this section mostly because I don't understand how this model works. It would be great if a more detailed discussion of the model was included. If limited in space this could also be thrown in the supplementary material of this paper. Correlations of this model to PMF do not look great and a discussion on the reasons why could be further investigated.*

The model simulates transport processes towards the Maïdo observatory without taking into account emissions. As such, we can qualify the sensitivity of in-situ measurements to surface emissions without taking the emission profiles or non-homogeneities in source distributions into account. These approximations result in the relatively bad correlation between PMF and SRRs.

*Section 3.3.1: How were the SRRs categorized in the model?*

The SRR categories are based on surface type (land vs water) and elevation (urban regions concentrated near the coast with surface elevation below 500 m above sea level).

*Line 526-530: The model and PMF have a weak correlation. Are these statements only based on Figure 15? What is the value of section 3.3.1 in the paper?*

We have completed the conclusion to include information in section 3.3.1, the identification of emission hotspots is indeed based on Figure 15.

*Figure 2, 5, 10: It would be a nice addition to generate monthly timeseries that would improve the discussions. Either the daily or monthly measurement figures could then be moved to the supplement. I think this will also improve the discussion of the figures since it is currently based on monthly trends.*

Monthly trends have been added in supplementary material.

*Figure 2: It would be great if the seasons could be added to the figure at the top and also the abbreviations mentioned in the caption. Many readers will first look at the figures and it would be easier to follow if the abbreviations are repeated in the caption. Also, I would strongly recommend that the authors change the x-axis to months instead of days of the year since the discussion is anyway referring to months.*

We have adapted the figure and captions.

**Technical comments referee #1:**

Thank you for your detailed remarks, we have adapted the manuscript as suggested.

**Specific comments referee #2:**

*As is understandable, the air at the remote high altitude is very clean compared to continental sites in Africa and Asia. This often poses challenges to the instrument detection limits and while the authors have generally done a good job in trying to address the issue, more details would be helpful for readers. For example, statistics on out of total number of measurements in each season for the reported compounds, how often were values below the detection limit. This could be provided as a table in the supplement to both inform about QA/QC and the challenges in measuring composition of clean air.*

We have added figures to illustrate the data quality (signal-to-noise ratio and data above limit of detection) in supplementary material.

*It is mentioned that calibration experiments were done frequently ..please provide the sensitivity in ncps/ppb of the compounds so that reader can see the drift of the 2 year deployment*

We have added the instantaneous calibration coefficients used throughout the measurement campaign in the supplementary material.

*Why were seasonal and diel profiles of six key VOC species analyzed which does not include acetone and acetaldehyde? This is sub-optimal use of the dataset.*

As we cannot go into detail on all 13 species monitored by the PTS-MS instrument, we opted to discuss compounds with clear emission sources to discuss seasonal and diel profiles. Other compounds are assumed to be implicitly discussed when addressing the contribution of each source identified by PMF and the chemical profile of these PMF source factors. Data for the other compounds however are available through the openly accessible dataset (Amelynck et al., 2021).

*What could be the reason for artefacts at m/z 93 which is normally well detected as toluene using a PTR-MS? On the other hand attribution of m/z 47 to ethanol and m/z 61 to formic acid and acetic acid is more challenging but no discussion of corrections and justification for these have been provided. What corrections were applied to correct for the HCHO back reaction with and its humidity dependence?*

HCHO is difficult to measure due to the back reaction but calibration coefficients were regularly measured versus air humidity. Nevertheless, comparison of our data against an Aerolaser instrument (based on Hantzsch derivatization) revealed PTR-MS HCHO nighttime (i.e. free troposphere at the Maïdo observatory) concentrations which were too high. This suggested potential interferences at m/z 31 from other compounds which could not be taken into account. Therefore, the HCHO data have not been considered.

The m/z 47 compounds were not considered in this paper as well because this ion signal was only regularly monitored from half 2018 onwards. Two main parent molecules resulting in this ion species are formic acid and ethanol (Baasandorj et al., 2015). Calibration of ethanol at m/z 47 by using a calibration gas unit provided by *Laboratoire des sciences du climat et de l'environnement* (*LSCE*, Climate and Environment Sciences Laboratory) showed a very low calibration coefficient for ethanol (0.33 ncps/ppbv) for our instrument. Additionally, huge ion signals during biomass plumes suggested that m/z 47 signals could mainly be attributed to formic acid (Verreyken et al., 2020). This information has been added to the manuscript.

Attribution of m/z 61 to acetic acid is indeed more challenging. According to Baasandorj et al. (2015), interfering compounds for acetic acid detection by PTR-MS with similar sensitivity as acetic acid are

glycolaldehyde, ethyl acetate and peroxyacetic acid. However, the strong co-variation of formic and acetic acid, which have similar sources and sinks in the atmosphere, suggest that the signal at m/z 61 could correspond mostly to acetic acid. We have included this in the new version of the manuscript.

*The calibration gas mixture used (e.g. Apel Riemer) typically also contains trimethyl benzene detected at m/z 121. Was this compound completely absent in ambient air?*

There was no trimethyl benzene present in the calibration mixture used.

*Was a mass scan ever done at some point during the 2 year deployment?*

Yes, we set out to focus on a selection of compounds, that were monitored sequentially by running the instrument in multiple ion detection model, but have occasionally performed mass scans, specifically during a period of time when the volcano was active on the island. It was found that the m/z ratios with significant data above the limit of detection were mostly included in the list of m/z ratios that were already selected beforehand.

*What was the residence time of air in the inlet and how often were filters changed during the 2 year long deployment?*

The residence time for the air masses to reach the PTR-MS instrument from the roof is estimated to be 3.2 s (2.5 s in the sampling line + 650 ms in the instrumental setup). On average, the filter was changed once every three weeks. This information has been added to the manuscript.

*Radiation has been used intensively in the analyses so more detailed description of radiation measurements should be provided in the methods section.*

Measurements of radiation have been made using a SPN1 Sunshine Pyranometer (Delta-T Devices Ltd., UK), stated accuracy 5% for both direct and diffuse radiation. This information has been added to the manuscript.

*It is mentioned that sugarcane is a major crop cultivated on the island . Is it known whether sugarcane waste is burnt in post-harvest seasons? If so this would be interesting to compare with the BB profile and literature reported emission factors of the measured compounds (see studies from FIREX campaign published in ACP special issue.*

The burning of agricultural waste in post-harvest season is illegal at La Réunion and it is thus uncommon on the island.

*The analyses of isoprene oxidation chemistry could benefit a lot from comparison with other studies from forested sites. What were the NO levels in summer when isoprene and Isoprene oxidation products are highest? It would be very valuable to compare the daytime ratio of MVK+MACR+Isop peroxides to isoprene to that observed from forested sites in Europe, Africa, Asia and South America. Comparing yields reported from laboratory studies such as Liu et al. Liu, Y. J., Herdlinger-Blatt, I., McKinney, K. A., and Martin, S. T.: Production of methyl vinyl ketone and methacrolein via the hydroperoxyl pathway of isoprene oxidation, Atmos. Chem. Phys., 13, 5715–5730, https://doi.org/10.5194/acp-13-5715-2013, 2013 would be useful too.*

We have added a figure on the ratio between isoprene and its oxidation products to the manuscript.

*Is there information about the boundary layer height at day and night in different seasons from the site? This would make the discussion about role of emissions and dilution more quantitative. In BB air masses what is the CO/CO2 mixing ratio?*

As the complex orographic profile leads to a highly variable boundary layer, large uncertainties are linked to boundary layer development at the location of Maïdo. Strong efforts were made to develop the FLEXPART-AROME Lagrangian transport model to reproduce turbulence in the planetary boundary layer in such a way that the transport model is consistent with the meteorological model. This was a significant improvement when modelling air mass transport towards the Maïdo observatory. However, the 2.5 x 2.5 km$^2$ model resolution of AROME is not able to correctly resolve local transport features which will affect the boundary layer development over the island. As such, we opt not to publish the PBL height locally at Maïdo from a meteorological model but refer to the typical mesoscale transport features and PBL development already described and published in literature and supplement it here using the FLEXPART-AROME backtrajectories. The SRRs shown in Figures 12 to 14 of the preprint illustrate the sensitivity of in-situ measurements at the location of Maïdo to surface emissions taking the planetary boundary layer movement into account.

The CO/CO2 ratio in BB air masses selected for the case studies presented in Verreyken et al. (2020) was within a range from 0.3 x 10$^{-3}$ to 0.5 x 10$^{-3}$.

*Detailed analyses of temperature and radiation regimes associated with highest isoprene emission and formation of photochemically formed compounds (see for e.g. Mishra et al. Emission drivers and variability of ambient isoprene, formaldehyde and acetaldehyde in north-west India during monsoon season, Environmental Pollution, Vol. 267, 115538, 2020) would also provide further mechanistic insights.*

We agree that detailed analyses of temperature and radiation regimes associated with highest isoprene emissions would be valuable. However, we only have measurements available at the location of Maïdo. Due to the complex profile, large variations due to the cloud-cover and temperature gradients with altitude, the meteorological circumstances at the Maïdo observatory are not necessarily reflective of the ones at the location where isoprene is emitted. Moreover, as this is a measurement report, we feel like the analysis as presented, and the improvements already implemented based on suggestions and comments of both referees, are sufficient for publication. We encourage others to make use of the openly available dataset for further analysis and will look into ways to exploit the dataset further ourselves.

*Back trajectory modelling description and analyses needs more clarity..can these be compared to local wind direction and wind rose too? Esp PMF factor profile wind roses would be helpful to supplement the other analyses.*

We have added daytime pollution roses for the different PMF source factors to the supplementary information. From these roses, the wind-separated behaviour is clear. No additional conclusions can be made from these plots.

*For PMF analyses would have been good to treat NOx as independent tracer with anthropogenic profile (Fig 11). Also for interpretation of anthropogenic source profile factor shown in Fig 11 would be really helpful to have the boundary layer height variation also in same plot even if from the met model in absence of measurements.*

We have added a figure showing the median diel profile separated according to wind-direction for NOx concentrations to the paper. This plot closely resembles the features of the anthropogenic tracer source factor contributions (enhanced from the west with a peak at the same time and the difference between western and eastern transport timed similarly) which were unique in the dataset.

*The conclusion that marine sources do not show up as source factor sounds strange for an island so authors should clarify this is so for the burden of the specific VOCs*

We have added this nuance to the conclusions.

**Minor points referee #2:**

We have addressed the technical comments brought up by referee #2 and below we only address the specific comments that were not addressed specifically in the new version of the manuscript.

*L6: Here and elsewhere should be sum of C8-aromatic compounds…also good to discuss which ones could be major contributors if speciation info from other studies is available*

During the OCTAVE intensive field campaign (7 March 2018 – 2 May 2018), speciation of $C_8$-aromatic compounds was done using GS-MS measurements which showed that on average the ratio between (m+p)-xylene:o-xylene:ethyl benzene was 57%:26%:18%. (Data provided by A. Colomb, personal communication). Data from the OCTAVE intensive field campaign is still under analysis and we do not think that including this information in this publication is suitable for this reason.

*L10: does not exhibit consistent diel variability is not clear… What is consistent diel variability? Authors should clarify*

We mean to say a typical diel pattern which is consistently present throughout the dataset. This has been clarified in the manuscript.

*L53: 2 years may not be adequate for inter-annual variability?*

Two years of data is indeed not sufficient for deriving annual trends. However, the dataset is sufficient to give indications on the variability among two years.

*L90: authors mention PBL variability but would be good to add info on the measured PBLs between day and night if known*

There are, as of yet, no measurements published showing PBL variation at the location of Maïdo. There were, however, such measurements performed sparsely on La Réunion throughout the period when the hs-PTR-MS was deployed. As this data does not cover a significant period of the measurement campaign, we opt not to publish it here. Instead, we refer to the typical mesoscale transport features and PBL development already described and published in literature. It is our opinion that this is sufficient for the current work.

*L106: How was m61 corrected for fragmentation effects?*

A dilute dynamic mixture of acetic acid in zero-VOC air was sampled by the hs-PTR-MS instrument to determine the transmission-corrected fractional contribution of the protonated acid ($f_{m/z61}$) and its fragments ($f_{m/z43}$) to the $H_3O^+$/acetic acid product ion distribution as a function of humidity, the latter being controlled by means of a dew point generator. The normalized ion signal at m/z 37 ($H_3O^+.H_2O$) was used as a proxy for the air humidity. The rate constant for the pathway leading to the protonated acetic acid (i.e. the product of the $H_3O^+$/$CH_3COOH$ collision rate constant and $f_{m/z61}$) was then further taken into account in the indirect calculation of $CH_3COOH$ mixing ratios.

*L108: Is is stated that m/z 137 was used for quantification and 81 was not, although at 136 Td one would expect more than 60% of MT signal to land at 81/ What fragmentation ratio was used and is there any information about the speciation of MT, even from other studies perhaps?*

Monoterpenes were quantified based on the ion signal at m/z 137, because it is considered as a better fingerprint for monoterpenes than the fragment at m/z 81, which can suffer from interferences from other compounds (e.g. hexenals and other $C_6$ compounds). As limonene was in the calibration bottle, the calibration coefficient of limonene at m/z 137 was taken into account for the monoterpene quantification. The $I_{137}/I_{81}$ ratio for limonene was 0.30. From GC-MS measurements performed during the OCTAVE intensive field campaign (7 March 2018 – 2 May 2018), it was found that limonene, alpha-pinene, and beta-pinene were the most abundant monoterpenes at the measurement site and the average limonene:alpha-pinene:beta-pinene ratio was 38%:52%:10% (A. Colomb, personal communication).

**References**

Baasandorj, M., et al.: Measuring Acetic and formic acid by proton-transfer-reaction mass-spectrometry: Sensitivity, humidity dependence, and quantifying interferences, Atmos. Meas. Tech., 8, 1303-1321, doi:10.5194/amt-8-1303-2015, 2015.

Rivera-Rios, J. C., et al. : Conversion of hydroperoxides to carbonyls in field and laboratory instrumentation : observational bias in diagnosing pristine versus anthropogenically controlled atmospheric chemistry, Geophys. Res. Let., 41, 8645-8651, doi:10.1002/2014GL061919, 2014.

Toon, O.B., et al.: Planning, implementation and scientific goals of the Studies of Emissions and Atmospheric Composition, Clouds and Climate Coupling by Regional Surveys (EEAC4RS) field mission, J. Geophys. Res. Atmos., 121, 4967-5009, doi:10.1002/2015JD024297, 2016.

Verreyken, B., et al.: Characterisation of African biomass burning plumes and impacts on the atmospheric composition over the south-west Indian Ocean, Atmos. Chem. Phys., 20, 14821-14845, doi:10.5194/acp-20-14821-2020, 2020.

Yáñez-Serrano, A. M., et al.: Atmospheric mixing ratios of methyl ethyl ketone (2-butanone) in tropical, boreal, temperate and marine environments, Atmos. Chem. Phys., 16, 10965–10984, doi:10.5194/acp-16-10965-2016, 2016.

Zhao J. and Zhang, R.: Proton transfer reaction rate constants between hydronium ion ($H_3O^+$) and volatile organic compounds, Atmos. Environ. 38, 2177-2185, 2004.